# Differentially Private Continual Release with Relative Error

**Bo Li** [1][2]  **Wei Wang** [1]  **Peng Ye** [1]

## Abstract

This work investigates several fundamental tasks, including MaxSum, MinSum, MaxSelect, and MinSelect, in the continual release model under differential privacy. Previous research has demonstrated that any algorithm for these tasks must admit a large purely *additive error*. We show that the error can be substantially reduced if a *relative error* term is allowed, provided that the input stream is generated *non-adaptively*. However, when input data records can be selected *adaptively*, we prove that a large error is inevitable for the task of selecting an attribute with a small cumulative sum, whereas small error bounds remain achievable for other tasks. This reveals a significant separation between non-adaptive and adaptive streams. We also complement our algorithms with nearly matching lower bounds.

## 1. Introduction

In today's data-driven world, the demand for privacy-preserving algorithms has become increasingly critical. As organizations gather and analyze vast amounts of personal data, ensuring individual privacy while still gaining valuable information poses a significant challenge. Differential privacy (DP) (Dwork et al., 2006b;a) has emerged as a leading framework for addressing this challenge, providing a rigorous mathematical definition of privacy that enables the release of useful information without compromising individual privacy.

Most research on differential privacy has focused on the *batch model*, in which a mechanism aggregates all input data and produces a single output. While this model effectively ensures privacy for static datasets, it fails to capture dynamic environments with continuously generated data. The *continual release* model (Dwork et al., 2010a; Chan et al., 2011) was proposed to overcome this limitation. In this model, data records arrive as a stream, and the mechanism is required to periodically update its output as new records are received, with privacy constraints being imposed on outputs over all time-steps.

The central challenge in the continual release model is that a single input record can influence outputs across multiple time-steps, enlarging the chance of privacy leakage. A substantial gap between the batch and continual release models was confirmed by Jain et al. (2023), who demonstrated that for certain fundamental tasks, transitioning from the batch model to the continual release model results in an unavoidable exponential increase in error.

The above hardness results hold for purely *additive error*. Motivated by the recent trend of studying differentially private algorithms with *relative error* guarantees, this work attempts to circumvent the barrier by incorporating an additional relative error term. In practice, relative error could be more meaningful than additive error since it scales with the magnitude of the true answer. When the true answer is extremely small, it could be completely overwhelmed by an additive error, leading to noninformative results. When the true answer is large, an additive error can be much smaller than it, rendering an overkill. In contrast, a relative error consistently offers an accurate estimate of the magnitude.

In many scenarios, input data records may depend on the algorithm's previous answers. For example, consider a platform that publishes real-time traffic conditions in a city. Drivers may decide their routes based on this information, thus affecting future traffic. This *adaptive* setting can be formalized by assuming an adversary who selects input records based on the algorithm's previous outputs. In this context, achieving accuracy is more challenging than in the *non-adaptive* setting, where the input stream is predetermined. However, Jain et al. (2023) established their upper bounds in the adaptive setting and their lower bounds in the non-adaptive setting. Therefore, their results do not reveal any difference between these two settings. They raised the question if it is possible to find a problem that distinguishes between non-adaptive and adaptive streams.

[1]Department of Computer Science and Engineering, The Hong Kong University of Science and Technology, Hong Kong SAR, China [2]Guangzhou HKUST Fok Ying Tung Research Institute, Guangzhou, China. Correspondence to: Peng Ye <pyeac@connect.ust.hk>.

*Proceedings of the 43rd International Conference on Machine Learning*, Seoul, South Korea. PMLR 306, 2026. Copyright 2026 by the author(s).

## 1.1. Results

In our setup, the input is a stream consisting of $T$ vectors in $[0, 1]^d$, which can be seen as $T$ records with $d$ attributes. The algorithm is required to estimate some statistics about the sum of the first $t$ vectors after receiving the $t$-th vector for every $t \in [T]$. In particular, we study the problems of approximately releasing and selecting the maximum across all attributes, denoted by MaxSum and MaxSelect. We also consider analogous problems for the minimum, referred to as MinSum and MinSelect. The formal definitions of these problems are provided in Section 2.2.

**Improved algorithms with relative error** We propose algorithms for these tasks (except MinSelect with adaptive inputs) that achieve $\mathrm{polylog}(d, T)$ additive error when a constant relative error is allowed, significantly improving over the optimal purely additive error established by Jain et al. (2023).[1] As a direct implication, our algorithms exhibit much better performance in the sparse setting where the true answer is $\mathrm{polylog}(d, T)$.

Our algorithms and analysis assume the input range is $[0, 1]$. This can be generalized to $[0, M]$ for any $M > 0$, in which case the final error bound scales by a factor of $M$. However, if negative inputs are allowed, our upper bound no longer apply. In fact, by adapting the proof strategy of Jain et al. (2023), one can show that the additive error cannot be improved even when relative error is considered. A more detailed explanation is provided in Appendix D.

**Nearly tight lower bounds** Alongside our algorithms, we provide nearly matching lower bounds. As a consequence, our results characterize the error for all these tasks up to poly-logarithmic factors. The error bounds for all tasks are summarized in Table 1.

**Separation between non-adaptive and adaptive streams** While a $\mathrm{polylog}(d, T)$ error is attainable for MinSelect with non-adaptive inputs, we prove that including a relative error does not yield improvements for adaptively selected input records—the best achievable error remains nearly the same as the purely additive error proved in (Jain et al., 2023). This highlights a strong separation between non-adaptive and adaptive streams in private continual release.

## 1.2. Technical Overview

Our algorithms for MaxSum, MinSum, and MaxSelect are based on a simple framework that updates the current answer once its error exceeds a certain threshold. We utilize

---

[1]Although Jain et al. (2023) only considered MaxSum and MaxSelect, these two problems are equivalent to MinSum and MinSelect for purely additive error, as replacing each record $x_i$ with $1 - x_i$ converts the minimum to the maximum.

the AboveThreshold mechanism to monitor if the error of the current output is within an acceptable range. If so, we keep the output unchanged, which incurs no privacy cost. Otherwise, we launch a batch algorithm on the received data to update the answer. Since the actual value increases by roughly $1 + \gamma$ after each update, we only need to perform around $\log_{1+\gamma} T \approx \log T / \gamma$ updates, leading to a $\mathrm{polylog}(d, T)$ error due to composition theorems.

Our algorithm for MinSelect follows a similar approach but requires a completely different utility analysis. After each update, we may switch to a new attribute with a smaller cumulative sum. Hence, we cannot guarantee that the minimum sum will increase by a multiplicative factor of $1 + \gamma$ after each update. This property makes it essentially different from other problems. To bound the number of updates, we exploit the fact that the stream is non-adaptive. Intuitively, an adversary attempting to attack the algorithm should increase the values for many coordinates because which coordinate will be chosen by the algorithm is unknown in advance. Thus, the minimum sum must increase by $1 + \gamma$ after a certain number of updates. This can be formally proved by a potential analysis inspired by an argument in (Asi et al., 2023a). We show that the number of updates is at most $\mathrm{polylog}(d, T)$, which yields a $\mathrm{polylog}(d, T)$ error.

Our lower bounds for the non-adaptive setting leverage the sequential embedding construction proposed by Jain et al. (2023). The key observation is that we can construct an input stream so that the sum along every coordinate is small. This makes the relative error term negligible compared to the additive error. Consequently, a lower bound for purely additive error translates to a lower bound on the additive error in our setup.

For MinSelect against an adaptive adversary, we establish a lower bound through a reduction from the problem of privately identifying the signs of consensus columns in a matrix. We transform the rows of the matrix into a stream of records so that the signs of consensus columns can be identified by locating attributes with zero sum, which can be solved by running an algorithm for MinSelect. Once such an attribute is found, we start to place 1's at this attribute (this operation makes the stream adaptively generated) and compel the algorithm to find a new attribute with zero sum. By repeating this procedure, we can determine the signs of most consensus columns. Therefore, existing hardness results for this problem imply lower bounds for MinSelect in the adaptive setting.

## 1.3. Related Work

The exploration of differentially private mechanisms in the continual release model began with the work of Dwork et al. (2010a) and Chan et al. (2011). They considered the fundamental problem of computing the sum of binary

*Table 1.* Nearly tight error bounds for tasks considered in this work. For readability, we omit $\text{polylog}(d, T, 1/\varepsilon, 1/\delta, 1/\gamma)$ terms. For MaxSum, MinSum, and MaxSelect, the bounds hold for both non-adaptive and adaptive streams. Also note that in order to cover all ranges of parameters, our results are combined with the additive-only upper bounds (Jain et al., 2023).

| TASK | PURE DP | APPROXIMATE DP | SOURCE |
|---|---|---|---|
| MaxSum AND MinSum | $\min\left\{\frac{1}{\gamma\varepsilon}, \frac{d}{\varepsilon}, \sqrt{\frac{T}{\varepsilon}}, T\right\}$ | $\min\left\{\frac{1}{\sqrt{\gamma}\varepsilon}, \frac{\sqrt{d}}{\varepsilon}, \frac{T^{1/3}}{\varepsilon^{2/3}}, T\right\}$ | THM. A.1, B.2 |
| MaxSelect | $\min\left\{\frac{1}{\gamma\varepsilon}, \frac{d}{\varepsilon}, \sqrt{\frac{T}{\varepsilon}}, T\right\}$ | $\min\left\{\frac{1}{\sqrt{\gamma}\varepsilon}, \frac{\sqrt{d}}{\varepsilon}, \frac{T^{1/3}}{\varepsilon^{2/3}}, T\right\}$ | THM. A.2, B.4 |
| MinSelect (NON-ADAPTIVE) | $\min\left\{\frac{1}{\gamma\varepsilon}, \frac{d}{\varepsilon}, \sqrt{\frac{T}{\varepsilon}}, T\right\}$ | $\min\left\{\frac{1}{\sqrt{\gamma}\varepsilon}, \frac{\sqrt{d}}{\varepsilon}, \frac{T^{1/3}}{\varepsilon^{2/3}}, T\right\}$ | THM. 3.1, B.4 |
| MinSelect (ADAPTIVE) | $\min\left\{\frac{d}{\varepsilon}, \sqrt{\frac{T}{\varepsilon}}, T\right\}$ | $\min\left\{\frac{\sqrt{d}}{\varepsilon}, \frac{T^{1/3}}{\varepsilon^{2/3}}, T\right\}$ | THM. 4.3 |

bits and proposed the celebrated binary tree mechanism whose error is only larger than that in the batch model by $\text{polylog}(T)$. Furthermore, an error of $\Omega(\log T)$ is shown to be inevitable (Dwork et al., 2010a), indicating that the continual release model is inherently more challenging than the batch model, in which an $O(1)$ error is attainable. A more pronounced separation was later discovered by Jain et al. (2023), who proved that for MaxSum and MaxSelect, the (purely additive) error in the continual release model can be exponentially larger than that in the batch model.

The aforementioned results focus solely on purely additive error. There has been a recent trend in the development of algorithms with relative accuracy guarantees. For various problems, such as frequency moments estimation and query release, it has been demonstrated that incorporating a relative error term can lead to substantial improvements (Epasto et al., 2023; Ghazi et al., 2023; 2025; Aamand et al., 2026).

The concept of differential privacy with adaptive input streams was formally defined by Jain et al. (2023), although it was also implicitly referenced in earlier works (e.g., (Smith & Thakurta, 2013)). It has been shown that non-adaptive and adaptive streams are equivalent under pure differential privacy, whereas a private algorithm in the non-adaptive setting may become non-private in the adaptive setting under approximate differential privacy (Denisov et al., 2022). In terms of utility, significant gaps between the two settings have been observed in private online learning (Asi et al., 2023b; Li et al., 2024). To our knowledge, such gaps have not yet been identified in the context of private continual release of statistics.

### 1.4. Organization

In Section 2, we provide the necessary background on differential privacy and define the problems we study. In Section 3, we present our algorithm for MinSelect in the non-adaptive setting and briefly analyze its accuracy. We establish our lower bound for MinSelect in the adaptive

setting in Section 4. Together, the results from these two sections emphasize our key finding—the separation between non-adaptive and adaptive streams. Algorithms and lower bounds for other problems, as well as proofs omitted from Sections 3 and 4, are deferred to the appendices.

## 2. Preliminaries

### 2.1. Differential Privacy

We start with the standard definition of differential privacy. Two datasets (or sequences) $S_0 = (x_1, \ldots, x_T)$ and $S_1 = (x'_1, \ldots, x'_T)$ of size $T$ over domain $\mathcal{X}$ are considered neighboring if they differ in at most one entry, i.e., there exists $j \in [T]$ such that $x_i = x'_i$ for all $i \in [T] \setminus \{j\}$. Differential privacy requires an algorithm's output distributions to be similar when running on neighboring datasets.

**Definition 2.1** (Differential Privacy (Dwork et al., 2006b;a))**.** A randomized algorithm $\mathcal{M}$ is $(\varepsilon, \delta)$-differentially private if for any pair of neighboring datasets $S_0, S_1 \in \mathcal{X}^T$ and any set $O$ of outputs, we have

$$\Pr[\mathcal{M}(S_0) \in O] \leq e^\varepsilon \Pr[\mathcal{M}(S_1) \in O] + \delta.$$

When $\delta = 0$, we often omit $\delta$ and say $\mathcal{M}$ is $\varepsilon$-differentially private. The case that $\delta > 0$ is referred to as approximate DP and the case that $\delta = 0$ is referred to as pure DP.

The above definition only applies to static datasets and does not capture the scenarios where data records can be adversarially generated. We next define differential privacy in the presence of adaptive inputs. Let $\mathcal{M}$ be a mechanism, $\texttt{Adv}$ be an adversary, and $b \in \{0, 1\}$ be a global parameter that is unknown to both $\mathcal{M}$ and $\texttt{Adv}$. The privacy game played between $\mathcal{M}$ and $\texttt{Adv}$ is defined as follows. At every round $t \in [T]$, the adversary $\texttt{Adv}$ generates two data points $x_t^{(0)}$ and $x_t^{(1)}$. The mechanism $\mathcal{M}$ receives $x_t^{(b)}$ and presents an output $a_t$ to $\texttt{Adv}$. The adversary $\texttt{Adv}$ has to ensure that $x_t^{(0)} = x_t^{(1)}$ for all $t \in [T] \setminus \{\tilde{t}\}$, where $\tilde{t}$ is a special round that is adaptively chosen by $\texttt{Adv}$ (i.e., the

choice can depend on the interaction before round $\tilde{t}$). For privacy, we require that it is hard for Adv to infer the value of $b$. Let $\mathsf{View}_{\mathcal{M},\mathtt{Adv}}(b)$ denote the view of Adv in this privacy game, which contains the internal randomness of Adv and $(a_1, \ldots, a_T)$ sent by $\mathcal{M}$. Differential privacy in the adaptive setting is defined as below.

**Definition 2.2** (Differential Privacy with Adaptive Inputs (Jain et al., 2023)). A randomized algorithm $\mathcal{M}$ is $(\varepsilon, \delta)$-differentially private in the adaptive continual observation model if for any adversary Adv and any set $O$ of views, we have

$$\Pr[\mathsf{View}_{\mathcal{M},\mathtt{Adv}}(0) \in O] \leq e^{\varepsilon} \Pr[\mathsf{View}_{\mathcal{M},\mathtt{Adv}}(1) \in O] + \delta.$$

Differential privacy enjoys the composition property, which allows us to design a complex private mechanism by combining multiple simple building blocks. Let $\mathcal{M}_1, \ldots, \mathcal{M}_k$ be $k$ mechanisms with independent randomness, denote by $\mathsf{Comp}(\mathcal{M}_1, \ldots, \mathcal{M}_k)$ a mechanism that interacts with Adv as follows. At round $t$, the adversary Adv generates $x_t^{(0)}$ and $x_t^{(1)}$. Then all mechanisms receive $x_t^{(b)}$ and respond $(a_{1,t}, \ldots, a_{k,t})$ to Adv. During each round, the responses $a_{1,t}, \ldots, a_{k,t}$ can be produced in arbitrary order and each response may depend on the previous ones. The following composition theorems ensure that $\mathsf{Comp}(\mathcal{M}_1, \ldots, \mathcal{M}_k)$ is private as long as every $\mathcal{M}_i$ is private (Dwork et al., 2006b; 2010b; Steinke, 2022; Vadhan & Wang, 2021; Lyu, 2022; Henzinger et al., 2026).

**Theorem 2.3** (Basic Composition). *Let $\mathcal{M}_1, \ldots, \mathcal{M}_k$ be $k$ randomized mechanisms with privacy parameters $(\varepsilon_1, \delta_1), \ldots, (\varepsilon_k, \delta_k)$. Then $\mathsf{Comp}(\mathcal{M}_1, \ldots, M_k)$ is $(\varepsilon, \delta)$-differentially private for $\varepsilon = \sum_{i=1}^{k} \varepsilon_i$ and $\delta = \sum_{i=1}^{k} \delta_i$.*

**Theorem 2.4** (Advanced Composition). *Let $\mathcal{M}_1, \ldots, \mathcal{M}_k$ be $k$ randomized mechanisms with privacy parameters $(\varepsilon_1, \delta_1), \ldots, (\varepsilon_k, \delta_k)$. Then $\mathsf{Comp}(\mathcal{M}_1, \ldots, M_k)$ is $(\varepsilon, \delta)$-differentially private for any $\delta > \sum_{j=1}^{k} \delta_j$ and $\varepsilon = \frac{1}{2} \sum_{j=1}^{k} \varepsilon_j^2 + \sqrt{2 \ln(1/\delta') \sum_{j=1}^{k} \varepsilon_j^2}$, where $\delta' = \delta - \sum_{j=1}^{k} \delta_j$.*

The following property of differential privacy, known as group privacy, quantifies the closeness between the output distributions obtained by running a private algorithm on two datasets that differ in multiple entries.

**Lemma 2.5** (Group Privacy). *If $\mathcal{M}$ is an $(\varepsilon, \delta)$-differentially private mechanism, then for any two datasets $S_0$ and $S_1$ that differ in $k$ entries and any event $O$, we have*

$$\Pr[\mathcal{M}(S_0) \in O] \leq e^{k\varepsilon} \Pr[\mathcal{M}(S_1) \in O] + \frac{e^{k\varepsilon} - 1}{e^{\varepsilon} - 1} \cdot \delta.$$

We then introduce some basic mechanisms for achieving differential privacy. The first is the Laplace mechanism.

**Definition 2.6** (Sensitivity). A function $f : \mathcal{X}^n \to \mathbb{R}$ has sensitivity $\Delta$ if $|f(S_0) - f(S_1)| \leq \Delta$ for any neighboring datasets $S_0$ and $S_1$.

**Theorem 2.7** (Laplace Mechanism (Dwork et al., 2006b)). *Suppose $f$ has sensitivity $\Delta$. The mechanism that outputs $f(S) + \mathrm{Lap}(\Delta/\varepsilon)$ is $\varepsilon$-differentially private. Moreover, we have $\Pr[|\mathrm{Lap}(\Delta/\varepsilon)| \leq \Delta \ln(1/\beta)/\varepsilon] \geq 1 - \beta$.*

The AboveThreshold mechanism, as depicted in Algorithm 1, allows for privately identifying the first query whose value exceeds a predefined threshold.

---

**Algorithm 1** AboveThreshold

---

**Input:** privacy parameter $\varepsilon$, threshold $\tau$, data stream $(x_1, \ldots, )$, query stream $(q_1, \ldots )$.
**Output:** stream $(\sigma_1, \ldots )$.
$\hat{\tau} \leftarrow \tau + \mathrm{Lap}(2/\varepsilon)$.
**for** $t = 1, \ldots,$ **do**
   $\nu_t \leftarrow \mathrm{Lap}(4/\varepsilon)$.
   **if** $q_t(x_1, \ldots, x_t) + \nu_t \geq \hat{\tau}$ **then**
      $\sigma_t \leftarrow \top$.
      **Halt**.
   **else**
      $\sigma_t \leftarrow \bot$.
   **end if**
**end for**

---

**Theorem 2.8** (Properties of AboveThreshold (Dwork et al., 2009; Dwork & Roth, 2014)). *The AboveThreshold mechanism (Algorithm 1) is an $\varepsilon$-differentially private algorithm that at round $t$ receives a data point $x_t \in \mathcal{X}$ and a sensitivity-1 query $q_t$ (may be chosen adaptively), and outputs $\sigma_t \in \{\top, \bot\}$. Moreover, suppose there are at most $k$ queries, then with probability $1 - \beta$ the AboveThreshold mechanism is $\alpha$-accurate for*

$$\alpha = \frac{8(\ln k + \ln(2/\beta))}{\varepsilon}.$$

*Namely, we have $q_t(x_1, \ldots, x_t) \geq \tau - \alpha$ for all $t$ with $\sigma_t = \top$, and $q_t(x_1, \ldots, x_t) \leq \tau + \alpha$ for all $t$ with $\sigma_t = \bot$.*

Let $H$ be a finite set and $\ell : \mathcal{X}^n \times H \to \mathbb{R}$ be a score function. We say $\ell$ has sensitivity $\Delta$ if $\ell(\cdot, h)$ has sensitivity $\Delta$ for any $h \in H$. Given a dataset $S$, the exponential mechanism outputs $h$ with probability

$$\frac{\exp(-\varepsilon \cdot \ell(S, h)/2\Delta)}{\sum_{f \in H} \exp(-\varepsilon \cdot \ell(S, f)/2\Delta)}.$$

As stated below, with high probability the exponential mechanism produces an $h \in H$ with low score.

**Theorem 2.9** (Exponential Mechanism (McSherry & Talwar, 2007)). *The exponential mechanism is $\varepsilon$-differentially*

*private. Moreover, with probability $1 - \beta$ it outputs an $h$ such that*

$$\ell(S, h) \leq \min_{f \in H} \ell(S, f) + \frac{2\Delta}{\varepsilon} \ln(|H|/\beta).$$

## 2.2. Problem Statement

Let $(x_1, \ldots, x_T)$ be a stream of length $T$, where every $x_i \in [0, 1]^d$. Given a function $f : \mathbb{R}^d \to \mathbb{R}$, we require an algorithm to accurately release $f(\sum_{s=1}^{t} x_s)$ at every round $t \in [T]$ while preserving $(\varepsilon, \delta)$-differential privacy.

We consider both estimation and selection tasks. For estimation, we say an algorithm is $(\gamma, \alpha, \beta)$-accurate for estimating $f$ if with probability at least $1 - \beta$, for every $t \in [T]$ the output $a_t$ satisfies

$$(1 - \gamma) \cdot f(\sum_{s=1}^{t} x_s) - \alpha \leq a_t \leq (1 + \gamma) \cdot f(\sum_{s=1}^{t} x_s) + \alpha.$$

In particular, we are interested in the cases that $f(x) = \max_{i \in [d]} x[i]$ and $f(x) = \min_{i \in [d]} x[i]$, where $x[i]$ denotes the value at the $i$-th coordinate of vector $x$. These two tasks are represented by MaxSum and MinSum.

We also study their selection variants. We say an algorithm is $(\gamma, \alpha, \beta)$-accurate for MaxSelect if with probability at least $1 - \beta$, for every $t \in [T]$ the output $a_t \in [d]$ satisfies

$$(1 - \gamma) \cdot \max_{i \in [d]} \sum_{s=1}^{t} x_s[i] - \alpha \leq \sum_{s=1}^{t} x_s[a_t].$$

Similarly, we say an algorithm is $(\gamma, \alpha, \beta)$-accurate for MinSelect if with probability at least $1 - \beta$, for every $t \in [T]$ the output $a_t \in [d]$ satisfies

$$\sum_{s=1}^{t} x_s[a_t] \leq (1 + \gamma) \cdot \min_{i \in [d]} \sum_{s=1}^{t} x_s[i] + \alpha.$$

## 3. Algorithm for MinSelect

In this section, we consider MinSelect with non-adaptive input streams. At each round, the mechanism is required to output an attribute with a cumulative sum that is not too larger than the minimum. In the batch model, this problem can be solved by the exponential mechanism.

We briefly describe our approach. At the beginning, we initialize the output by sampling uniformly from $[d]$. At each round $t \in [T]$, we check if the output remains accurate using the AboveThreshold mechanism. Once the current output becomes inaccurate, we launch an exponential mechanism to update it.

The final error bound depends on the privacy parameter of the exponential mechanism and the AboveThreshold mechanism, which, by the composition property of DP, is determined by the number of times the output is updated.

If the output is updated frequently, we should set a small privacy parameter for the exponential mechanism and the AboveThreshold mechanism, leading to a large error. In contrast, if the output is updated rarely, we can use a larger privacy parameter, which results in a smaller error.

Therefore, the central challenge is to bound the number of updates. For other tasks (i.e., MaxSum, MinSum, and MaxSelect), this can be easily done by leveraging the fact that the true answer is multiplied by $1 + \Theta(\gamma)$ after each update. Since the sum is at most $T$, the number of updates cannot exceed $O(\log_{1+\Theta(\gamma)} T) = O(\log T / \gamma)$. However, this simple argument becomes invalid for MinSelect since it is possible that we switch to a new coordinate with an even smaller sum after an update.

To fix this issue, we will prove that with high probability, the minimum sum still grows by a fraction of $\Theta(\gamma)$ after $\text{polylog}(d, T)$ updates. As a result, the total number of updates is at most $O(\log T / \gamma) \cdot \text{polylog}(d, T)$ and we obtain a $\text{polylog}(d, T)/\gamma$ error bound. Due to some reasons that will be explained later, we should gradually decrease the privacy parameter as the minimum sum grows. This is ensured by decreasing the privacy parameter after every $K = \text{polylog}(d, T)$ updates. We illustrate the detailed implementation in Algorithm 2 and state the final results in the following theorem.

**Theorem 3.1.** *Let $d, T \in \mathbb{N}$, $\varepsilon, \gamma, \beta \in (0, 1)$, and $\delta \in [0, 1/2)$. In the non-adaptive continual release model, Algorithm 2 is $(\varepsilon, \delta)$-differentially private and $(\gamma, \alpha, \beta)$-accurate for MinSelect, where*

- $\alpha = O\left( \frac{\log(dT/\gamma\beta)}{\gamma} \cdot \frac{\log(dT/\beta)}{\varepsilon} \right)$ *if $\delta = 0$.*

- $\alpha = O\left( \sqrt{\frac{\log(dT/\gamma\beta)\log(1/\delta)}{\gamma}} \cdot \frac{\log(dT/\beta)}{\varepsilon} \right)$ *if $\delta > 0$.*

Here, we only present the core idea of our analysis and defer the formal proof to Appendix C.1. Based on the value of $i$ in Algorithm 2, we can divide the algorithm into several stages. Suppose that we enter the $i$-th stage at some round $t_1$, define a potential

$$\Phi_t = \sum_{j \in [d]} \exp(-\varepsilon_i \ell_t(j)/2)$$

for $t \geq t_1$, where $\ell_t$ is as in Algorithm 2. Since the stream is generated non-adaptively, the potential at every round is fixed until we move forward to the next stage.

Let $M = \min_{j \in [d]} \ell_{t_1}(j)$ and $t_2$ be a time-step such that $\min_{j \in [d]} \ell_{t_2}(j) \approx (1 + \gamma/2)M + \alpha/2$. We will prove that the number of updates before $t_2$ is at most $K = \text{polylog}(d, T)$ with high probability. Therefore, after every $K$ updates, the minimum sum is increased by a

**Algorithm 2** Privately Releasing MinSelect

---

**Input:** time horizon $T \in \mathbb{N}$, privacy parameters $\varepsilon, \delta$, relative error parameter $\gamma$, failure probability $\beta$, data stream $(x_1, \ldots, x_T)$, where $x_i \in [0, 1]^d$.

**Output:** stream $(a_1, \ldots, a_T) \in [d]^T$.

$K \leftarrow \lceil 11(\ln d + 8(\ln T + \ln(8dT/\beta))) + 40 \ln(20/\gamma\beta) + 40 \ln(4 \ln T/(\ln(1 + \gamma/3)\beta)) \rceil$.

**if** $\delta = 0$ **then**

    $\alpha \leftarrow \frac{896K(\ln T + \ln(8dT/\beta))}{\varepsilon\gamma}$.

**else**

    $\alpha \leftarrow 8\sqrt{\frac{3K \ln(1/\delta)}{\gamma}} \cdot \frac{64(\ln T + \ln(8dT/\beta))}{\varepsilon}$.

**end if**

$\varepsilon_j \leftarrow \frac{64(\ln T + \ln(8dT/\beta))}{\alpha}$ for $j \leq \lceil 4/\gamma \rceil$ and $\varepsilon_j \leftarrow \frac{32(\ln T + \ln(8dT/\beta))}{\alpha \cdot (1 + \gamma/3)^{j-1-\lceil 4/\gamma \rceil}}$ for $j > \lceil 4/\gamma \rceil$.

$\alpha_j \leftarrow \frac{8(\ln T + \ln(8T/\beta))}{\varepsilon_j}$ for all $j$.

$a_0 \leftarrow k$ where $k$ is sampled uniformly from $[d]$.

$i \leftarrow 1$ and $i' \leftarrow 1$.

Initiate an instance of AboveThreshold with privacy parameter $\varepsilon_1$ and threshold 0.

**for** $t = 1, \ldots, T$ **do**

    Define query $q_t \equiv -(1 + \gamma) \cdot \min_{j \in [d]} \sum_{s=1}^t x_s[j] - \alpha + \sum_{s=1}^t x_s[a_{t-1}] + \alpha_i$.

    Feed $q_t$ to AboveThreshold and obtain outcome $\sigma_t$.

    **if** $\sigma_t = \top$ **then**

        Define $\ell_t(j) \equiv \sum_{s=1}^t x_s[j]$ for $j \in [d]$.

        Set $a_t \leftarrow k$ with probability $\frac{\exp(-\varepsilon_i \ell_t(k)/2)}{\sum_{j \in [d]} \exp(-\varepsilon_i \ell_t(j)/2)}$.

        $i' \leftarrow i' + 1$ and $i \leftarrow \lceil i'/K \rceil$.

        Initiate an instance of AboveThreshold with privacy parameter $\varepsilon_i$ and threshold 0.

    **else**

        $a_t \leftarrow a_{t-1}$.

    **end if**

**end for**

---

multiplicative factor of $1 + \gamma/2$ (and an additive factor of $\alpha/2$). The total number of updates can then be bounded by $\mathrm{polylog}(d, T)$.

Suppose that the algorithm has made an update at round $t \in [t_1, t_2]$. Let $t' \in [t_1 + 1, t_2]$ be some round such that $\Phi_{t'} \approx \Phi_t/e$. The probability that it will make the next update before round $t'$ is at most (we assume that the AboveThreshold mechanism has zero error for simplicity, which will be handled by adjusting the coefficients in the formal proof)

$$P \approx \sum_{j \in [d]} \frac{\exp(-\varepsilon_0 \ell_t(j)/2)}{\Phi_t} \cdot \mathbb{I}[\ell_{t'}(j) > (1 + \gamma)M + \alpha].$$

We can prove that $P$ is bounded by some constant that is strictly less than 1. Intuitively, the output distribution of the exponential mechanism at round $t$ should concentrate on those $j$'s with $\ell_t(j) \leq (1 + \gamma/2)M + \alpha/2$. If $P$ is very close to 1, we must have $\ell_{t'}(j) > (1 + \gamma)M + \alpha$ for most of them. However, this will cause a dramatic drop in potential, contradicting that $\Phi_{t'} \approx \Phi_t/e$. As a consequence, the number of updates during $[t + 1, t']$ can be bounded by a geometric random variable with mean

$$P + P^2 + \cdots = P/(1 - P) = O(1).$$

We now consider the entire time interval $[t_1, t_2]$. It is not difficult to see from the definition that $\Phi_{t_1} \leq d \exp(-\varepsilon_i M/2)$ and $\Phi_{t_2} \geq \exp(-\varepsilon_i((1 + \gamma/2)M + \alpha/2)/2)$. As a consequence, we can split the time interval $[t_1 + 1, t_2]$ into $r$ phases such that the potential drops by $e$ during each phase, where $r = \ln(\Phi_{t_1}/\Phi_{t_2}) \leq \ln d + \varepsilon_i(\gamma M + \alpha)/4$.

In our formal proof, we show that our choice of $\varepsilon_i$ satisfies $\varepsilon_i(\gamma M + \alpha) = \mathrm{polylog}(d, T)$. Therefore, we have $r = \mathrm{polylog}(d, T)$. By standard concentration arguments, the number of updates before round $t_2$ is at most $K = \mathrm{polylog}(d, T)$. Since $M$ (the minimum sum among all attributes) grows gradually, we must decrease $\varepsilon_i$ accordingly to ensure that $\varepsilon_i(\gamma M + \alpha) = \mathrm{polylog}(d, T)$ always holds.

## 4. Adaptive Lower Bound for MinSelect

In this section, we prove our lower bound for MinSelect in the adaptive setting. Our approach is to show that we can solve the problem of privately determining the signs of consensus columns in a matrix by interacting with a private algorithm for MinSelect. This problem was also used by Talwar et al. (2015) and Asi et al. (2023b) to prove lower bounds for other problems.

The following result says that any $(\varepsilon, \delta)$-differentially private algorithm solving this problem requires $\tilde{\Omega}(\sqrt{d}/\varepsilon)$ samples. Note that the version used by Talwar et al. (2015) and Asi et al. (2023b) is stated for $\varepsilon = 1$. Here we extend it to other values of $\varepsilon$ and provide a proof in Appendix C.2 for completeness. In this paper, when the input is a matrix, differential privacy is defined by viewing each row of the matrix as an input item.

**Lemma 4.1.** *Let $d$ be sufficiently large and $\varepsilon \leq 0.1$. For some $n = \Theta(\frac{\sqrt{d}}{\varepsilon \log d})$, suppose that $\mathcal{A}$ is an algorithm that takes as input a matrix $X \in \{-1, 1\}^{n \times d}$ and outputs a $d$-dimensional vector. Denote by $W$ the set of indices of consensus columns in $X$. If*

$$\Pr_{\mathcal{A}} \left[ \sum_{j \in W} \mathbb{I}[\mathcal{A}(X)[j] = X_{1,j}] \geq 3d/4 \right] \geq 2/3$$

*given that $|W| \geq 0.999d$. Then $\mathcal{A}$ is not $(\varepsilon, \delta)$-differentially private for $\delta = o(1/n^3)$.*

For pure DP, we will use the following lemma, which can be proved by the classical packing argument (Hardt & Talwar,

2010). We enclose a proof in Appendix C.3 for completeness.

**Lemma 4.2.** *Let $\mathcal{A}$ be an $\varepsilon$-differentially private algorithm that takes as input a matrix $X \in \{-1, 1\}^{n \times d}$ and outputs a $d$-dimensional vector such that*

$$\Pr_{\mathcal{A}} \left[ \sum_{j \in [d]} \mathbb{I}[\mathcal{A}(X)[j] = X_{1,j}] \geq 4d/5 \right] \geq 2/3$$

*given that all columns in $X$ are consensus columns. Then $n = \Omega(d/\varepsilon)$.*

We illustrate the idea of our proof for sufficiently large $T$. Let $\mathcal{A}$ be an algorithm for MinSelect against an adaptive adversary. We show that we can use $\mathcal{A}$ to solve the problem of identifying the signs of the columns.

Given an input matrix $X$, we create a new matrix by converting each of its columns into two columns. If an entry in $X$ is $-1$, we place a pair $(0, 1)$ in the same row of the two corresponding columns. If it is $1$, we put a $(1, 0)$ instead. By doing so, every consensus column in $X$ is converted into two consensus columns, and the index of the all-zero column reflects the sign of the original column.

We then send the rows of the new matrix as data records to $\mathcal{A}$. Now each column becomes an attribute in the data stream. Let $\alpha$ denote the additive error of $\mathcal{A}$. Since there is an all-zero column, algorithm $\mathcal{A}$ will output the index of an attribute whose sum is at most $\alpha$. After that, we start to put 1's at this attribute. This increases the sum along this attribute and forces $\mathcal{A}$ to change its output to other attributes.

If $\alpha$ is less than the number of rows of the matrix, then $\mathcal{A}$ will never select an attribute associated with an all-one column as long as there is an unselected all-zero attribute. Under the condition that most columns in $X$ are consensus columns, we can identify the signs of a large fraction of them by repeating the above procedure. As a consequence, a lower bound on the sample complexity of identifying the signs of columns yields a lower bound on $\alpha$.

Applying a case analysis that was also used in (Jain et al., 2023), we can prove the following lower bound that covers all ranges of parameters. It holds for any $\gamma \geq 0$ and matches the additive-only upper bound from (Jain et al., 2023).

**Theorem 4.3.** *Let $T, d \in \mathbb{N}$ be sufficiently large. Any $(\varepsilon, \delta)$-differentially private mechanism $\mathcal{M}$ solving* MinSelect *in the adaptive continual release model with relative error $\gamma$ and success probability $2/3$ must incur an additive error of*

- $\alpha = \Omega \left( \min \left\{ d/\varepsilon, \sqrt{T/\varepsilon}, T \right\} \right)$ *if $\delta = 0$.*

- $\alpha = \tilde{\Omega} \left( \min \left\{ \frac{\sqrt{d}}{\varepsilon}, \frac{T^{1/3}}{\varepsilon^{2/3}}, T \right\} \right)$ *if $\delta > 0$ and $\delta = o(\varepsilon/T^3)$.*

*Proof.* To begin with, note that the lower bound must be a non-decreasing function of $d$. This is because any algorithm for MinSelect with dimension $d^* > d$ can be used to solve MinSelect with dimension $d$ by padding each input record with $d^* - d$ ones. If the output falls in the extra $d^* - d$ coordinates, we can simply output one of the $d$ coordinates, as this can only make the sum smaller.

We then prove that $\alpha = \Omega(T)$ when $\varepsilon T \leq 2$. Let $S$ be a sequence with $T/4$ $(0, 1)$'s followed by $3T/4$ zero vectors and $S'$ be a sequence with $T/4$ $(1, 0)$'s followed by $3T/4$ zero vectors. Let $a_T$ and $a_T'$ be the outputs of $\mathcal{M}$ at round $T$ when running on $S$ and $S'$, respectively. Suppose $\alpha \leq T/9$, we have $\Pr[a_T = 1] \geq 2/3$, which implies $\Pr[a_T' \neq 1] \leq \sqrt{e} \cdot \Pr[a_T \neq 1] + 2\delta/\varepsilon < 2/3$. However, we also have $\Pr[a_T' \neq 1] = \Pr[a_T' = 2] \geq 2/3$, a contradiction. Thus, we can conclude that $\alpha > T/9 = \Omega(T)$.

From now on, we assume $\varepsilon > 2/T$. We first focus on the case that $\delta > 0$. Consider the matrix $X \in \{-1, 1\}^{n \times d/2}$ in Lemma 4.1, where $n = \Theta \left( \frac{\sqrt{d}}{\varepsilon \log d} \right)$. Suppose $d \leq O((\varepsilon T)^{2/3})$, we have $T \geq 2nd/5$. For any $t \in [n]$, create $x_t \in \{0, 1\}^d$ as follows:

- $x_t[2j - 1] = \frac{X_{t,j} + 1}{2}$ for all $j \in [d/2]$.

- $x_t[2j] = \frac{-X_{t,j} + 1}{2}$ for all $j \in [d/2]$.

We construct an algorithm $\mathcal{A}$ that takes $X$ as input and privately estimates the signs of consensus columns in $X$ by interacting with $\mathcal{M}$. It first feeds $(x_1, \dots, x_n)$ to $\mathcal{M}$. Then, for every $r \in [2d/5 - 1]$ it streams $(x_{rn+1}, \dots, x_{rn+n}) = (e_{a_{rn}}, \dots, e_{a_{rn}})$ to $\mathcal{M}$, where $a_{rn}$ is the output of $\mathcal{M}$ at round $rn$ and $e_{a_{rn}}$ is the $a_{rn}$-th standard basis vector in $\mathbb{R}^d$ (i.e., $e_{a_{rn}}$ is a $d$-dimensional vector with 1 in the $a_{rn}$-th coordinate and 0 elsewhere). Finally, it outputs a vector such that

$$\mathcal{A}(X)[j] = 2\mathbb{I}[2j \in \{a_n, \dots, a_{(2d/5)n}\}] - 1$$

for every $j \in [d/2]$.

We show that $\mathcal{A}$ is $(\varepsilon, \delta)$-differentially private. Let $X_0$ and $X_1$ be two matrices that differ in one row. Consider an adversary Adv that generates two sequences $(x_1^{(0)}, \dots, x_n^{(0)})$ and $(x_1^{(1)}, \dots, x_n^{(1)})$ from $X_0$ and $X_1$ using the aforementioned construction, respectively. At round $t \in [n]$, the vector $x_t^{(b)}$ is sent to $\mathcal{M}$, where $b \in \{0, 1\}$ is unknown to Adv. Then, for every $r \in [2d/5 - 1]$ and $i \in [n]$ the adversary sends $e_{a_{rn}}$ to $\mathcal{M}$ at round $t = rn + i$, where $a_{rn}$ is the output of $\mathcal{M}$ at round $rn$. Observe that $\mathcal{A}(X_b)$ can be seen as a post-processing of $\text{View}_{\mathcal{M}, \text{Adv}}(b)$. The fact that $\mathcal{M}$ is $(\varepsilon, \delta)$-differentially private for adaptive streams implies that $\mathcal{A}$ is $(\varepsilon, \delta)$-differentially private.

Let $W$ be the set of consensus columns in $X$ and assume that $|W| \geq 0.999d/2$. Note that for each consensus column

$j \in W$, either $x_1[2j-1] = \cdots = x_n[2j-1] = 0$ or $x_1[2j] = \cdots = x_n[2j] = 0$. Since $0.999d/2 > 2d/5$, by our construction, there exists $k^* \in [d]$ such that $x_1[k^*] = \cdots = x_{T'}[k^*] = 0$, where $T' = 2nd/5 \leq T$. As a consequence, with probability $2/3$ the mechanism $\mathcal{M}$ selects $a_t$ such that

$$\sum_{s=1}^{t} x_s[a_t] \leq (1+\gamma) \cdot 0 + \alpha = \alpha$$

for every $t \in [T']$.

We will prove by contradiction that $\alpha \geq n$. Indeed, if $\alpha < n$, by our construction we know that $a_n, a_{2n}, \ldots, a_{(2d/5)n}$ are distinct since

$$\sum_{t=1}^{r'n} x_t[a_{rn}] \geq n > \alpha$$

for any $r < r'$. Let $S_b = \{k \in [d] : x_1[k] = \cdots = x_n[k] = b\}$ for $b \in \{0, 1\}$, we have $0.999d/2 \leq |S_b| \leq d/2$. Note that $\{a_n, a_{2n}, \ldots, a_{(2d/5)n}\} \cap S_1 = \emptyset$. Therefore,

$$
\begin{aligned}
&|\{a_n, a_{2n}, \ldots, a_{(2d/5)n}\} \cap S_0| \\
=&2d/5 - |\{a_n, a_{2n}, \ldots, a_{(2d/5)n}\} \cap S_1| \\
&- |\{a_n, a_{2n}, \ldots, a_{(2d/5)n}\} \cap ([d] \setminus (S_0 \cup S_1))| \\
\geq&2d/5 - 0.001d \\
=&0.399d.
\end{aligned}
$$

Pick any $j \in W$. If $X_{1,j} = 1$, we have $2j \in S_0$ and

$$\mathbb{I}[\mathcal{A}(X)[j] \neq X_{1,j}] = \mathbb{I}[2j \notin \{a_n, a_{2n}, \ldots, a_{(2d/5)n}\}].$$

If $X_{1,j} = -1$, we have $2j - 1 \in S_0$. This implies $2j \in S_1$. Since $\{a_n, a_{2n}, \ldots, a_{(2d/5)n}\} \cap S_1 = \emptyset$, it must be the case that $2j \notin \{a_n, a_{2n}, \ldots, a_{(2d/5)n}\}$. Therefore, we always have $\mathcal{A}(X)[j] = -1 = X_{1,j}$. The error of $\mathcal{A}$ can then be bounded by

$$
\begin{aligned}
&\sum_{j \in W} \mathbb{I}[\mathcal{A}(X)[j] \neq X_{1,j}] \\
=&\sum_{j \in W} \mathbb{I}[2j \in S_0 \setminus \{a_n, a_{2n}, \ldots, a_{(2d/5)n}\}] \\
\leq&\left| S_0 \setminus \{a_n, a_{2n}, \ldots, a_{(2d/5)n}\} \right| \\
=&|S_0| - \left| S_0 \cap \{a_n, a_{2n}, \ldots, a_{(2d/5)n}\} \right| \\
\leq&d/2 - 0.399d \\
=&0.101d.
\end{aligned}
$$

As a consequence, we have

$$
\begin{aligned}
&\sum_{j \in W} \mathbb{I}[\mathcal{A}(X)[j] = X_{1,j}] \\
=&|W| - \sum_{j \in W} \mathbb{I}[\mathcal{A}(X)[j] \neq X_{1,j}] \\
\geq&0.999d/2 - 0.101d \\
=&0.3985d.
\end{aligned}
$$

This contradicts Lemma 4.1, which suggests that the above cannot exceed $3/4 \cdot (d/2) = 0.375d$. Hence, we have $\alpha \geq n = \Omega\left(\frac{\sqrt{d}}{\varepsilon \log d}\right)$.

When $d \geq \Omega((\varepsilon T)^{2/3})$, by the fact that the lower bound is non-decreasing, we have

$$\alpha = \Omega\left(\frac{\sqrt{(\varepsilon T)^{2/3}}}{\varepsilon \log((\varepsilon T)^{2/3})}\right) = \tilde{\Omega}\left(\frac{T^{1/3}}{\varepsilon^{2/3}}\right).$$

Now, we move on to the case that $\delta = 0$. Similarly, consider a matrix $X \in \{-1, 1\}^{n \times d/2}$ whose columns are all consensus columns, where $n = \lceil \alpha + 1 \rceil$. Suppose $d \leq \sqrt{\varepsilon T}$. If $T < 2nd/5$, we have $\alpha = \Omega(T/d) = \Omega(\sqrt{T/\varepsilon})$. Otherwise, we can construct an algorithm $\mathcal{A}$ in the same way and use the same argument to show that

$$
\begin{aligned}
\sum_{j \in [d/2]} \mathbb{I}[\mathcal{A}(X)[j] = X_{1,j}] &\geq d/2 - (d/2 - 2d/5) \\
&= 2d/5 \\
&\geq 4/5 \cdot (d/2).
\end{aligned}
$$

By Lemma 4.2, we have $\alpha = \Omega(d/\varepsilon)$. The $\Omega(\sqrt{T/\varepsilon})$ bound for $d > \sqrt{\varepsilon T}$ follows by the fact that the lower bound is non-decreasing in $d$ and setting $d = \lfloor \sqrt{\varepsilon T} \rfloor$. $\qquad \square$

# 5. Conclusion

This paper investigates four fundamental tasks (MaxSum, MinSum, MaxSelect, and MinSelect) in the continual release model under differential privacy. In contrast to the purely additive error guarantees considered by previous research, we study algorithms with both relative and additive error.

We develop algorithms with poly-logarithmic additive error for MaxSum, MinSum, and MaxSelect with adaptive streams, and for MinSelect with non-adaptive streams, significantly improving existing purely additive error bounds. In addition, we provide nearly matching lower bounds, showing the optimality of our proposed algorithms. For MinSelect with adaptive streams, we prove that incorporating a relative error cannot reduce the additive error. This demonstrates a notable separation between non-adaptive and adaptive streams in private continual release.

There remain poly-logarithmic gaps between our upper and lower bounds. Closing these gaps is a direct future direction. Another research direction is to explore if similar improvements can be made by involving a relative error for other tasks in differentially private continual release.

## Acknowledgements

The research was supported in part by an NSFC grant 62432008, RGC RIF grant R6021-20, an RGC TRS grant T43-513/23N-2, RGC CRF grants C7004-22G, C1029-22G and C6015-23G, NSFC/RGC grant CRS_HKUST601/24 and RGC GRF grants 16207922, 16207423, 16203824 and 16211123.

## Impact Statement

This paper presents work whose goal is to advance the field of machine learning. There are many potential societal consequences of our work, none of which we feel must be specifically highlighted here.

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

## A. Algorithms

### A.1. Algorithm for MaxSum and MinSum

We provide a general algorithm that privately and continually estimates any function $f$ of the cumulative sums that is non-negative, monotone, and has bounded sensitivity. As a direct consequence, this solves MaxSum and MinSum.

The algorithm keeps the output unchanged until its error is larger than our desired bound. This is achieved by the AboveThreshold mechanism. Once the output becomes invalid, we update it by adding Laplace noise to the correct answer. In addition, we decrease the privacy parameter ($\varepsilon_i$'s) used in the Laplace mechanism and the AboveThreshold mechanism after each update. This helps us save a factor of $\log T$ in the final error. Note that the error remains poly-logarithmic even if we use a consistent privacy parameter throughout the execution. This is in contrast to Algorithm 2, in which a decreasing privacy parameter is necessary for achieving a poly-logarithmic error.

We describe the details in Algorithm 3. The final error bound is stated in Theorem A.1.

---

**Algorithm 3** Privately Releasing $f$

---

**Input:** time horizon $T \in \mathbb{N}$, privacy parameters $\varepsilon, \delta$, relative error parameter $\gamma$, failure probability $\beta$, function $f : \mathbb{R}_{\geq 0}^d \to \mathbb{R}_{\geq 0}$, data stream $(x_1, \ldots, x_T)$, where $x_i \in [0, 1]^d$.
**Output:** stream $(a_1, \ldots, a_T) \in \mathbb{R}_{\geq 0}^T$.
**if** $\delta = 0$ **then**
    $\alpha \leftarrow \frac{40}{\varepsilon\gamma} \cdot 8(\ln T + \ln(4T/\beta))$.
**else**
    $\alpha \leftarrow \frac{64(\ln T + \ln(4T/\beta))}{\epsilon} \cdot \sqrt{\frac{42\ln(1/\delta)}{\gamma}}$.
**end if**
$\varepsilon_j \leftarrow \frac{6}{\alpha} \cdot 8(\ln T + \ln(4T/\beta))$ for $j \leq \lceil 1/\gamma \rceil$ and $\varepsilon_j \leftarrow \frac{2}{\alpha} \cdot \frac{8(\ln T + \ln(4T/\beta))}{(1+\gamma/3)^{j-\lceil 1/\gamma \rceil-1}}$ for $j > \lceil 1/\gamma \rceil$.
$\alpha_j \leftarrow \frac{8(\ln T + \ln(4T/\beta))}{\varepsilon_j}$ for all $j$.
$a_0 \leftarrow (1 + \gamma) \cdot f(\vec{0}) + \alpha$.
$i \leftarrow 1$.
Initiate an instance of AboveThreshold with privacy parameter $\varepsilon_1$ and threshold 0.
**for** $t = 1, \ldots, T$ **do**
    Define query $q_t \equiv (1 - \gamma) \cdot f(\sum_{s=1}^t x_s) - \alpha - a_{t-1} + \alpha_i$.
    Feed $q_t$ to AboveThreshold and obtain outcome $\sigma_t$.
    **if** $\sigma_t = \top$ **then**
        $a_t \leftarrow (1 + 2\gamma/3) \cdot f(\sum_{s=1}^t x_s) + 2\alpha/3 + \mathrm{Lap}(2/\varepsilon_i)$.
        $i \leftarrow i + 1$.
        Initiate an instance of AboveThreshold with privacy parameter $\varepsilon_i$ and threshold 0.
    **else**
        $a_t \leftarrow a_{t-1}$.
    **end if**
**end for**

---

**Theorem A.1.** *Let $d, T \in \mathbb{N}$, $\varepsilon, \gamma, \beta \in (0, 1)$, $\delta \in [0, 1/2)$, and $f : \mathbb{R}_{\geq 0}^d \to \mathbb{R}_{\geq 0}$ be a function such that $0 \leq f(x + y) - f(x) \leq 1$ for any $x \in \mathbb{R}_{\geq 0}^d$ and $y \in [0, 1]^d$. In the adaptive continual release model, Algorithm 3 is $(\varepsilon, \delta)$-differentially private and $(\gamma, \alpha, \beta)$-accurate for estimating $f$, where*

- $\alpha = O\left(\frac{\log(T/\beta)}{\varepsilon\gamma}\right)$ *if $\delta = 0$.*

- $\alpha = O\left(\frac{\log(T/\beta)}{\varepsilon} \cdot \sqrt{\frac{\log(1/\delta)}{\gamma}}\right)$ *if $\delta > 0$.*

*Proof.* We start by proving the privacy of Algorithm 3. The entire algorithm is composed of a series of AboveThreshold algorithms and Laplace mechanisms, where the $i$-th execution of each is $\varepsilon_i$-differentially private. By basic and advanced

composition theorems, it suffices to verify $\sum_{i=1}^{T} \varepsilon_i \leq \varepsilon/2$ for $\delta = 0$ and $\frac{1}{2} \sum_{i=1}^{T} \varepsilon_i^2 + \sqrt{2\ln(1/\delta) \sum_{i=1}^{T} \varepsilon_i^2} \leq \varepsilon/2$ for $\delta > 0$. For the former, we have

$$
\begin{aligned}
\sum_{i=1}^{T} \varepsilon_i &\leq \sum_{i=1}^{\lceil 1/\gamma \rceil} \varepsilon_i + \sum_{i > \lceil 1/\gamma \rceil} \varepsilon_i \\
&\leq \frac{6}{\alpha} \cdot 8(\ln T + \ln(4T/\beta)) \cdot \lceil 1/\gamma \rceil + \frac{2}{\alpha} \cdot 8(\ln T + \ln(4T/\beta)) \sum_{j=0}^{\infty} \frac{1}{(1 + \gamma/3)^j} \\
&= \frac{6}{\alpha} \cdot 8(\ln T + \ln(4T/\beta)) \cdot \lceil 1/\gamma \rceil + \frac{2}{\alpha} \cdot 8(\ln T + \ln(4T/\beta)) \cdot \frac{3 + \gamma}{\gamma} \\
&\leq \frac{6}{\alpha} \cdot 8(\ln T + \ln(4T/\beta)) \cdot \frac{2}{\gamma} + \frac{2}{\alpha} \cdot 8(\ln T + \ln(4T/\beta)) \cdot \frac{4}{\gamma} \\
&\leq \varepsilon/2.
\end{aligned}
$$

For the latter, we have

$$
\begin{aligned}
\sum_{i=1}^{T} \varepsilon_i^2 &\leq \sum_{i=1}^{\lceil 1/\gamma \rceil} \varepsilon_i^2 + \sum_{i > \lceil 1/\gamma \rceil} \varepsilon_i^2 \\
&\leq \left( \frac{6}{\alpha} \cdot 8(\ln T + \ln(4T/\beta)) \right)^2 \cdot \lceil 1/\gamma \rceil + \left( \frac{2}{\alpha} \cdot 8(\ln T + \ln(4T/\beta)) \right)^2 \sum_{j=0}^{\infty} \frac{1}{(1 + \gamma/3)^{2j}} \\
&\leq \left( \frac{8(\ln T + \ln(4T/\beta))}{\alpha} \right)^2 \cdot (72/\gamma + 12/\gamma) \\
&\leq \frac{\varepsilon^2}{32 \ln(1/\delta)}.
\end{aligned}
$$

The above is less than 1, hence

$$
\begin{aligned}
\frac{1}{2} \sum_{i=1}^{T} \varepsilon_i^2 + \sqrt{2\ln(1/\delta) \sum_{i=1}^{T} \varepsilon_i^2} &\leq \frac{1}{2} \sqrt{\sum_{i=1}^{T} \varepsilon_i^2} + \sqrt{2\ln(1/\delta) \sum_{i=1}^{T} \varepsilon_i^2} \\
&\leq 2\sqrt{2\ln(1/\delta) \sum_{i=1}^{T} \varepsilon_i^2} \\
&\leq \varepsilon/2.
\end{aligned}
$$

We then prove the utility. By Theorem 2.8, with probability $1 - \beta/2T$ the $i$-th execution of AboveThreshold is $\alpha_i$-accurate. By Theorem 2.7, with probability $1 - \beta/2T$ the $i$-th Laplace mechanism has error no more than $2\ln(2T/\beta)/\varepsilon_i$. From now on, we condition on the event that all the above happen. By the union bound, this holds with probability $1 - \beta$.

Let $t_1 < t_2 < \cdots$ be the time-steps at which AboveThreshold responses $\top$. For $i \leq \lceil 1/\gamma \rceil$, we have $2\ln(2T/\beta)/\varepsilon_i \leq \alpha/3$. Therefore,

$$
\begin{aligned}
a_{t_i} &\leq (1 + 2\gamma/3) \cdot f(\sum_{s=1}^{t_i} x_s) + 2\alpha/3 + 2\ln(2T/\beta)/\varepsilon_i \\
&\leq (1 + \gamma) \cdot f(\sum_{s=1}^{t_i} x_s) + \alpha
\end{aligned}
$$

and

$$a_{t_i} \geq (1 + 2\gamma/3) \cdot f(\sum_{s=1}^{t_i} x_s) + 2\alpha/3 - 2\ln(2T/\beta)/\varepsilon_i$$

$$\geq (1 + 2\gamma/3) \cdot f(\sum_{s=1}^{t_i} x_s) + \alpha/3.$$

Moreover, $\sigma_{t_i} = \top$ implies $q_{t_i} \geq -\alpha_i$, which yields

$$f(\sum_{s=1}^{t_i} x_s) \geq (1 - \gamma) \cdot f(\sum_{s=1}^{t_i} x_s)$$

$$\geq a_{t_i-1} - 2\alpha_i + \alpha$$

$$\geq a_{t_i-1} + 2\alpha/3$$

since $2\alpha_i \leq \alpha/3$. Using the above results and $a_0 \geq \alpha$, it can be shown by induction that $f(\sum_{s=1}^{t_i} x_s) \geq i \cdot \alpha$ for $i \leq \lceil 1/\gamma \rceil$. We then again prove by induction that

$$f(\sum_{s=1}^{t_i} x_s) \geq \frac{\alpha}{\gamma} \cdot (1 + \gamma/3)^{i-\lceil 1/\gamma \rceil}$$

for $i \geq \lceil 1/\gamma \rceil$. Note that this already holds for $i = \lceil 1/\gamma \rceil$. Suppose that it holds for some $i \geq \lceil 1/\gamma \rceil$, we have

$$a_{t_i} \geq (1 + 2\gamma/3) \cdot f(\sum_{s=1}^{t_i} x_s) + 2\alpha/3 - 2\ln(2T/\beta)/\varepsilon_i$$

$$\geq (1 + \gamma/3) \cdot f(\sum_{s=1}^{t_i} x_s)$$

since

$$2\ln(2T/\beta)/\varepsilon_i \leq \frac{\alpha}{3} \cdot (1 + \gamma/3)^{i-\lceil 1/\gamma \rceil}$$

$$\leq \frac{\gamma}{3} \cdot f(\sum_{s=1}^{t_i} x_s).$$

Then, $q_{t_{i+1}} \geq -\alpha_{i+1}$ implies

$$f(\sum_{s=1}^{t_{i+1}} x_s) \geq (1 - \gamma) \cdot f(\sum_{s=1}^{t_{i+1}} x_s) + \gamma \cdot f(\sum_{s=1}^{t_i} x_s)$$

$$\geq \alpha + a_{t_i} - 2\alpha_{i+1} + \gamma \cdot f(\sum_{s=1}^{t_i} x_s)$$

$$\geq (1 + \gamma/3) \cdot f(\sum_{s=1}^{t_i} x_s)$$

since

$$2\alpha_{i+1} \leq \alpha \cdot (1 + \gamma/3)^{i-\lceil 1/\gamma \rceil}$$

$$\leq \gamma \cdot f(\sum_{s=1}^{t_i} x_s).$$

As a consequence, for any $i > \lceil 1/\gamma \rceil$, we also have

$$a_{t_i} \leq (1 + 2\gamma/3) \cdot f(\sum_{s=1}^{t_i} x_s) + 2\alpha/3 + 2\ln(2T/\beta)/\varepsilon_i$$

$$\leq (1 + \gamma) \cdot f(\sum_{s=1}^{t_i} x_s) + \alpha.$$

By the monotonicity of $f$, this directly extends to $a_t \leq (1 + \gamma) \cdot f(\sum_{s=1}^{t} x_s) + \alpha$ for all $t \in [T]$.

Finally, we prove a lower bound on $a_t$. We have already shown

$$a_{t_i} \geq (1 + 2\gamma/3) \cdot f(\sum_{s=1}^{t_i} x_s) + \alpha/3$$

$$\geq (1 - \gamma) \cdot f(\sum_{s=1}^{t_i} x_s) - \alpha$$

for $i \leq \lceil 1/\gamma \rceil$ and

$$a_{t_i} \geq (1 + \gamma/3) \cdot f(\sum_{s=1}^{t_i} x_s)$$

$$\geq (1 - \gamma) \cdot f(\sum_{s=1}^{t_i} x_s) - \alpha$$

for $i > \lceil 1/\gamma \rceil$. For $t \notin \{t_1, t_2, \dots\}$, let $i = \min\{j : t < t_j\}$. Since $\sigma_t = \bot$, we have $q_t \leq \alpha_i$, which directly leads to

$$a_t = a_{t-1}$$

$$\geq (1 - \gamma) \cdot f(\sum_{s=1}^{t_i} x_s) - \alpha + \alpha_i - \alpha_i$$

$$= (1 - \gamma) \cdot f(\sum_{s=1}^{t_i} x_s) - \alpha.$$

$\square$

## A.2. Algorithm for MaxSelect

Our algorithm for MaxSelect is similar to Algorithm 3, with the Laplace mechanism replaced by the exponential mechanism. We illustrate the details in Algorithm 4 and the results in Theorem A.2.

**Theorem A.2.** *Let $d, T \in \mathbb{N}$, $\varepsilon, \gamma, \beta \in (0, 1)$, and $\delta \in [0, 1/2)$. In the adaptive continual release model, Algorithm 4 is $(\varepsilon, \delta)$-differentially private and $(\gamma, \alpha, \beta)$-accurate for MaxSelect, where*

- $\alpha = O\left(\frac{\log(dT/\beta)}{\varepsilon\gamma}\right)$ *if $\delta = 0$.*

- $\alpha = O\left(\frac{\log(dT/\beta)}{\varepsilon} \cdot \sqrt{\frac{\log(1/\delta)}{\gamma}}\right)$ *if $\delta > 0$.*

*Proof.* The proof strategy is similar to the proof of Theorem A.1. For privacy, by basic and advanced composition theorems, it suffices to verify $\sum_{i=1}^{T} \varepsilon_i \leq \varepsilon/2$ for $\delta = 0$ or $\frac{1}{2}\sum_{i=1}^{T} \varepsilon_i^2 + \sqrt{2\ln(1/\delta)\sum_{i=1}^{T}\varepsilon_i^2} \leq \varepsilon/2$ for $\delta > 0$. For the former, we

---

**Algorithm 4** Privately Releasing MaxSelect

---

**Input:** time horizon $T \in \mathbb{N}$, privacy parameters $\varepsilon, \delta$, relative error parameter $\gamma$, failure probability $\beta$, data stream $(x_1, \ldots, x_T)$, where $x_i \in [0, 1]^d$.
**Output:** stream $(a_1, \ldots, a_T) \in [d]^T$.
**if** $\delta = 0$ **then**
   $\alpha \leftarrow \frac{768(\ln T + \ln(4dT/\beta))}{\varepsilon\gamma}$.
**else**
   $\alpha \leftarrow \sqrt{\frac{224\ln(1/\delta)}{\gamma}} \cdot \frac{48(\ln T + \ln(4dT/\beta))}{\varepsilon}$.
**end if**
$\varepsilon_j \leftarrow \frac{48(\ln T + \ln(4dT/\beta))}{\alpha}$ for $j \leq \lceil 3/\gamma \rceil$ and $\varepsilon_j \leftarrow \frac{48(\ln T + \ln(4dT/\beta))}{\alpha \cdot (1+\gamma/3)^{j - \lceil 3/\gamma \rceil - 1}}$ for $j > \lceil 3/\gamma \rceil$.
$\alpha_j \leftarrow \frac{8(\ln T + \ln(4T/\beta))}{\varepsilon_j}$ for all $j$.
$a_0 \leftarrow 1$.
$i \leftarrow 1$.
Initiate an instance of AboveThreshold with privacy parameter $\varepsilon_1$ and threshold 0.
**for** $t = 1, \ldots, T$ **do**
   Define query $q_t \equiv (1 - \gamma) \cdot \max_{j \in [d]} \sum_{s=1}^{t} x_s[j] - \alpha - \sum_{s=1}^{t} x_s[a_{t-1}] + \alpha_i$.
   Feed $q_t$ to AboveThreshold and obtain outcome $\sigma_t$.
   **if** $\sigma_t = \top$ **then**
      Define $\ell_t(j) \equiv -\sum_{s=1}^{t} x_s[j]$ for $j \in [d]$.
      Set $a_t \leftarrow k$ with probability $\frac{\exp(-\varepsilon_i \ell_t(k)/2)}{\sum_{j \in [d]} \exp(-\varepsilon_i \ell_t(j)/2)}$.
      $i \leftarrow i + 1$.
      Initiate an instance of AboveThreshold with privacy parameter $\varepsilon_i$ and threshold 0.
   **else**
      $a_t \leftarrow a_{t-1}$.
   **end if**
**end for**

---

have

$$\sum_{i=1}^{T} \varepsilon_i \leq \sum_{i=1}^{\lceil 3/\gamma \rceil} \varepsilon_i + \sum_{i > \lceil 3/\gamma \rceil} \varepsilon_i$$

$$\leq \frac{48(\ln T + \ln(4dT/\beta))}{\alpha} \cdot \lceil 3/\gamma \rceil + \frac{48(\ln T + \ln(4dT/\beta))}{\alpha} \sum_{j=0}^{\infty} \frac{1}{(1 + \gamma/3)^j}$$

$$\leq \frac{48(\ln T + \ln(4dT/\beta))}{\alpha} \cdot \frac{4}{\gamma} + \frac{48(\ln T + \ln(4dT/\beta))}{\alpha} \cdot \frac{4}{\gamma}$$

$$\leq \varepsilon/2.$$

For the latter, we have

$$\sum_{i=1}^{T} \varepsilon_i^2 \leq \sum_{i=1}^{\lceil 3/\gamma \rceil} \varepsilon_i^2 + \sum_{i > \lceil 3/\gamma \rceil} \varepsilon_i^2$$

$$\leq \left( \frac{48(\ln T + \ln(4dT/\beta))}{\alpha} \right)^2 \cdot \lceil 3/\gamma \rceil + \left( \frac{48(\ln T + \ln(4dT/\beta))}{\alpha} \right)^2 \sum_{j=0}^{\infty} \frac{1}{(1 + \gamma/3)^{2j}}$$

$$\leq \left( \frac{48(\ln T + \ln(4dT/\beta))}{\alpha} \right)^2 \cdot (4/\gamma + 3/\gamma)$$

$$\leq \frac{\varepsilon^2}{32\ln(1/\delta)}.$$

The above is less than 1, hence

$$\frac{1}{2}\sum_{i=1}^{T}\varepsilon_i^2 + \sqrt{2\ln(1/\delta)\sum_{i=1}^{T}\varepsilon_i^2} \leq \frac{1}{2}\sqrt{\sum_{i=1}^{T}\varepsilon_i^2} + \sqrt{2\ln(1/\delta)\sum_{i=1}^{T}\varepsilon_i^2}$$

$$\leq 2\sqrt{2\ln(1/\delta)\sum_{i=1}^{T}\varepsilon_i^2}$$

$$\leq \varepsilon/2.$$

We next prove the utility guarantee. By Theorem 2.8, with probability $1 - \beta/2T$ the $i$-th execution of AboveThreshold is $\alpha_i$-accurate. By Theorem 2.9, with probability $1 - \beta/2T$ the $i$-th exponential mechanism has error no more than $2\ln(2dT/\beta)$. From now on, we condition on the event that all the above happen. By the union bound, this holds with probability $1 - \beta$.

Let $t_1 < t_2 < \cdots$ be the time-steps in which AboveThreshold responses $\top$. For $i \leq \lceil 3/\gamma \rceil$, we have $2\ln(2dT/\beta)/\varepsilon_i \leq \alpha/3$. Therefore,

$$\sum_{s=1}^{t_i} x_s[a_{t_i}] \geq \max_{j\in[d]}\sum_{s=1}^{t_i} x_s[j] - \alpha/3.$$

Moreover, $\sigma_{t_i} = \top$ implies $q_{t_i} \geq -\alpha_i$, which yields

$$\max_{j\in[d]}\sum_{s=1}^{t_i} x_s[j] \geq (1-\gamma)\cdot\max_{j\in[d]}\sum_{s=1}^{t_i} x_s[j]$$

$$\geq \sum_{s=1}^{t_i} x_s[a_{t_{i-1}}] + \alpha - 2\alpha_i$$

$$\geq \sum_{s=1}^{t_i} x_s[a_{t_{i-1}}] + 2\alpha/3$$

since $2\alpha_i \leq \alpha/3$. Using the above results, it can be shown by induction that

$$\max_{j\in[d]}\sum_{s=1}^{t_i} x_s[j] \geq \frac{i+1}{3}\cdot\alpha$$

for $i \leq \lceil 3/\gamma \rceil$.

We then again prove by induction that

$$\max_{j\in[d]}\sum_{s=1}^{t_i} x_s[j] \geq \frac{\alpha}{\gamma}\cdot(1+\gamma/3)^{i-\lceil 3/\gamma \rceil}$$

for $i \geq \lceil 3/\gamma \rceil$. Note that this already holds for $i = \lceil 3/\gamma \rceil$. Suppose that it holds for some $i \geq \lceil 3/\gamma \rceil$, we have

$$\sum_{s=1}^{t_i} x_s[a_{t_i}] \geq \max_{j\in[d]}\sum_{s=1}^{t_i} x_s[j] - 2\ln(2dT/\beta)/\varepsilon_i$$

$$\geq (1-\gamma/3)\cdot\max_{j\in[d]}\sum_{s=1}^{t_i} x_s[j]$$

since

$$2\ln(2dT/\beta)/\varepsilon_i \leq \frac{\alpha}{3}\cdot(1+\gamma/3)^{i-\lceil 3/\gamma \rceil}$$

$$\leq \frac{\gamma}{3}\cdot\max_{j\in[d]}\sum_{s=1}^{t_i} x_s[j].$$

Then, $q_{t_{i+1}} \geq -\alpha_{i+1}$ implies

$$(1 - 2\gamma/3) \cdot \max_{j \in [d]} \sum_{s=1}^{t_{i+1}} x_s[j] = (1 - \gamma) \cdot \max_{j \in [d]} \sum_{s=1}^{t_{i+1}} x_s[j] + \gamma/3 \cdot \max_{j \in [d]} \sum_{s=1}^{t_{i+1}} x_s[j]$$

$$\geq \sum_{s=1}^{t_{i+1}} x_s[a_{t_i}] + \alpha - 2\alpha_{i+1} + \gamma/3 \cdot \max_{j \in [d]} \sum_{s=1}^{t_{i+1}} x_s[j]$$

$$\geq \sum_{s=1}^{t_{i+1}} x_s[a_{t_i}]$$

since

$$2\alpha_{i+1} \leq \alpha/3 \cdot (1 + \gamma/3)^{i - \lceil 3/\gamma \rceil}$$

$$\leq \gamma/3 \cdot \max_{j \in [d]} \sum_{s=1}^{t_i} x_s[j]$$

$$\leq \gamma/3 \cdot \max_{j \in [d]} \sum_{s=1}^{t_{i+1}} x_s[j].$$

Thus, we have

$$\max_{j \in [d]} \sum_{s=1}^{t_{i+1}} x_s[j] \geq \frac{1}{1 - 2\gamma/3} \sum_{s=1}^{t_{i+1}} x_s[a_{t_i}]$$

$$\geq \frac{1}{1 - 2\gamma/3} \sum_{s=1}^{t_i} x_s[a_{t_i}]$$

$$\geq \frac{1 - \gamma/3}{1 - 2\gamma/3} \max_{j \in [d]} \sum_{s=1}^{t_i} x_s[j]$$

$$\geq \frac{\alpha}{\gamma} \cdot (1 + \gamma/3)^{i+1 - \lceil 3/\gamma \rceil},$$

where the last inequality is due to the inductive hypothesis and $\frac{1 - \gamma/3}{1 - 2\gamma/3} \geq (1 + \gamma/3)$.

We now prove the desired result. For $i \leq \lceil 3/\gamma \rceil$, we already have

$$\sum_{s=1}^{t_i} x_s[a_{t_i}] \geq \max_{j \in [d]} \sum_{s=1}^{t_i} x_s[j] - \alpha/3$$

$$\geq (1 - \gamma) \cdot \max_{j \in [d]} \sum_{s=1}^{t_i} x_s[j] - \alpha.$$

For $i > \lceil 3/\gamma \rceil$, we also have

$$\sum_{s=1}^{t_i} x_s[a_{t_i}] \geq (1 - \gamma/3) \cdot \max_{j \in [d]} \sum_{s=1}^{t_i} x_s[j]$$

$$\geq (1 - \gamma) \cdot \max_{j \in [d]} \sum_{s=1}^{t_i} x_s[j] - \alpha.$$

Now consider $t \notin \{t_1, t_2, \dots\}$ and let $i = \min\{j : t < t_j\}$. Since $\sigma_t = \bot$, we have $q_t \le \alpha_i$, which directly leads to

$$\sum_{s=1}^{t} x_s[a_t] = \sum_{s=1}^{t} x_s[a_{t-1}]$$

$$\ge (1-\gamma) \cdot \max_{j \in [d]} \sum_{s=1}^{t_i} x_s[j] - \alpha + \alpha_i - \alpha_i$$

$$= (1-\gamma) \cdot \max_{j \in [d]} \sum_{s=1}^{t_i} x_s[j] - \alpha.$$

$\square$

# B. Non-Adaptive Lower Bounds

## B.1. Non-Adaptive Lower Bound for MaxSum and MinSum

Following the proof strategy of Jain et al. (2023), we show that an algorithm for MaxSum or MinSum can be used to release one-way marginals. We will make use of the following hardness result.

**Lemma B.1** ((Hardt & Talwar, 2010; Lyu & Talwar, 2025)). *Let $d$ be sufficiently large and $\mathcal{X} = \{0,1\}^d$. Suppose $\mathcal{A}$ is an algorithm that takes as input a dataset $S = (x_1, \dots, x_n) \in \mathcal{X}^n$ and outputs a $d$-dimensional vector such that*

$$\Pr_{\mathcal{A}} \left[ \left| \mathcal{A}(S)[j] - \sum_{i=1}^{n} x_i[j] \right| \le n/100 \text{ for all } j \in [d] \right] \ge 2/3.$$

*Then:*

- *For some $n = \Theta(d/\varepsilon)$, algorithm $\mathcal{A}$ cannot be $\varepsilon$-differentially private.*

- *For some $n = \Theta(\sqrt{d}/\varepsilon)$, algorithm $\mathcal{A}$ cannot be $(\varepsilon, o(1/n))$-differentially private.*

**Theorem B.2.** *Let $d, T \in \mathbb{N}$ be sufficiently large. Any $(\varepsilon, \delta)$-differentially private mechanism $\mathcal{M}$ solving MaxSum or MinSum in the non-adaptive continual release model with relative error $\gamma$ and success probability $2/3$ must incur an additive error of*

- $\alpha = \Omega\left(\min\left\{\frac{1}{\varepsilon\gamma}, \frac{d}{\varepsilon}, \sqrt{\frac{T}{\varepsilon}}, T\right\}\right)$ *if $\delta = 0$.*

- $\alpha = \Omega\left(\min\left\{\frac{\sqrt{1/\gamma}}{\varepsilon}, \frac{\sqrt{d}}{\varepsilon}, \frac{T^{1/3}}{\varepsilon^{2/3}}, T\right\}\right)$ *if $\delta > 0$ and $\delta = o(\varepsilon/T)$.*

*Proof.* Assume $\gamma \le 1/1000$. Similar to the argument used in (Jain et al., 2023), we can prove that $\alpha = \Omega(T)$ when $\varepsilon T \le 2$. Note that the lower bound should be a non-decreasing function of $d$, since an algorithm for $d$-dimensional data can be used to solve the problem for $d^*$-dimensional data ($d^* < d$) by padding each input vector with $d - d^*$ zeros. Therefore, it suffices to consider $d = 1$. Let $S$ be a sequence with $T$ 0's and $S'$ be another sequence with $3T/4$ 0's followed by $T/4$ 1's. Let $a_T$ and $a'_T$ be the output of $\mathcal{M}$ at round $T$ on $S$ and $S'$, respectively. Suppose $\alpha \le T/9$, we have $\Pr[a_T \le T/9] \ge 2/3$. By group privacy, we have $\Pr[a'_T > T/9] \le \sqrt{e} \cdot \Pr[a_T > T/9] + 2\delta/\varepsilon < 2/3$ for sufficiently large $T$. But by the utility of $\mathcal{M}$, we also have $\Pr[a'_T > T/9] > \Pr[a'_T > (1-\gamma)T/4 - \alpha] \ge 2/3$, a contradiction. Thus, we have $\alpha > T/9 = \Omega(T)$.

From now on, we assume that $\varepsilon > 2/T$. We first consider MaxSum. Let $S = (x_1, \dots, x_n)$ be a dataset, where $x_i \in \{0,1\}^d$. Let $e_i$ be the $i$-th standard basis vector in $\mathbb{R}^d$ and $\overline{e_i} = 1 - e_i$. We construct a stream $(y_1, \dots, y_{T'})$ for $T' = 2nd$ as follows:

- $y_t = x_t$ for $t \in [n]$.

- $y_{n+2n(i-1)+1} = \cdots = y_{n+2n(i-1)+n} = e_i$ for all $i \in [d]$.

- $y_{n+2n(i-1)+n+1} = \cdots = y_{n+2ni} = \overline{e_i}$ for all $i \in [d-1]$.

Note that at round $t = n + 2n(i-1) + n$ $(i \in [d])$, we have

$$\sum_{s=1}^{t} y_s[i] = \sum_{k=1}^{n} x_k[i] + in$$

and

$$\sum_{s=1}^{t} y_s[j] = \sum_{k=1}^{n} x_k[j] + (i-1)n$$

for all $j \neq i$. Thus,

$$\sum_{s=1}^{t} y_s[i] \geq \sum_{s=1}^{t} y_s[j]$$

for all $j \neq i$. If $T \geq T'$ and we run $\mathcal{M}$ on $(y_1, \ldots, y_{T'})$, then with probability $2/3$, the output $a_t$ should satisfy

$$a_t \leq (1+\gamma) \sum_{s=1}^{t} y_s[i] + \alpha$$

$$= (1+\gamma) \sum_{k=1}^{n} x_k[i] + (1+\gamma)in + \alpha$$

$$\leq \sum_{k=1}^{n} x_k[i] + \gamma n + (1+\gamma)in + \alpha$$

and

$$a_t \geq (1-\gamma) \sum_{s=1}^{t} y_s[i] - \alpha$$

$$= (1-\gamma) \sum_{k=1}^{n} x_k[i] + (1-\gamma)in - \alpha$$

$$\geq \sum_{k=1}^{n} x_k[i] - \gamma n + (1-\gamma)in - \alpha.$$

Hence, $a_t - in$ is an estimate of $\sum_{k=1}^{n} x_k[i]$ with absolute error

$$\gamma(i+1)n + \alpha \leq \gamma(d+1)n + \alpha.$$

Suppose $d \leq \lfloor 1/1000\gamma \rfloor$. The above error is at most $\gamma(1/1000\gamma + 1)n + \alpha \leq n/500 + \alpha$.

For the case that $\delta = 0$, we take $n = \Theta(d/\varepsilon)$ as in Lemma B.1. When $d \leq O(\sqrt{\varepsilon T})$, we have $T \geq 2nd = T'$. Hence, we can use $\mathcal{M}$ to construct an $(\varepsilon, \delta)$-differentially private algorithm to release the one-way marginals with error $n/500 + \alpha$. This implies $\alpha \geq n/100 - n/500 = \Omega(d/\varepsilon)$ because otherwise it will contradict Lemma B.1. When $d \geq \Omega(\sqrt{\varepsilon T})$, taking $d^* = \Theta(\sqrt{\varepsilon T})$ gives $\alpha = \Omega(d^*/\varepsilon) = \Omega(\sqrt{T/\varepsilon})$ since the lower bound is non-decreasing in $d$.

The case that $\delta > 0$ is similar. We set $n = \Theta(\sqrt{d}/\varepsilon)$. When $d \leq O((\varepsilon T)^{2/3})$, we have $T \geq 2nd = T'$. By the same argument we have $\alpha = \Omega(\sqrt{d}/\varepsilon)$. When $d \geq \Omega((\varepsilon T)^{2/3})$, taking $d^* = \Theta((\varepsilon T)^{2/3})$ yields $\alpha = \Omega(\sqrt{d^*}/\varepsilon) = \Omega(T^{1/3}/\varepsilon^{2/3})$.

It remains to prove the lower bound for $d > \lfloor 1/1000\gamma \rfloor$. We can again take $d^* = \lfloor 1/1000\gamma \rfloor$, which gives

$$\alpha = \Omega\left(\min\left\{d^*/\varepsilon, \sqrt{T/\varepsilon}\right\}\right) = \Omega\left(\min\left\{1/\varepsilon\gamma, \sqrt{T/\varepsilon}\right\}\right)$$

for $\delta = 0$ and

$$\alpha = \Omega\left(\min\left\{\sqrt{d^*}/\varepsilon, T^{1/3}/\varepsilon^{2/3}\right\}\right) = \Omega\left(\min\left\{1/\varepsilon\sqrt{\gamma}, T^{1/3}/\varepsilon^{2/3}\right\}\right)$$

for $\delta > 0$. Summarizing all cases yield our desired lower bound.

The proof for MinSum is analogous. We just flip the construction of the input stream, i.e., we set

- $y_t = x_t$ for $t \in [n]$.

- $y_{n+2n(i-1)+1} = \cdots = y_{n+2n(i-1)+n} = \overline{e_i}$ for all $i \in [d]$.

- $y_{n+2n(i-1)+n+1} = \cdots = y_{n+2ni} = e_i$ for all $i \in [d-1]$.

We can similarly show that $a_t - (i-1)n$ is an estimate of $\sum_{k=1}^{n} x_k[i]$ with error $\gamma dn + \alpha$ for $t = n + 2n(i-1) + n$. The result then follows by an identical argument. $\square$

### B.2. Non-Adaptive Lower Bound for MaxSelect and MinSelect

Our proof is based on the construction of Jain et al. (2023). The idea is to show that an algorithm for MaxSelect or MinSelect in the continual release model can be used to simultaneously solve multiple instances of the same problem in the batch model. Therefore, the desired lower bound can be proved by leveraging the following lemma.

**Lemma B.3** ((Jain et al., 2023)). *Let $\mathcal{X} = \{0,1\}^d$, where $d = d'k$. Suppose $\mathcal{M}$ is an $(\varepsilon, \delta)$-differentially private algorithm that takes as input a dataset $S = \{x_1, \ldots, x_n\} \in \mathcal{X}^n$ and outputs a k-dimensional vector such that*

$$\mathcal{M}(S)[i] \in \{(i-1)d' + 1, \ldots, id'\} \text{ for all } i \in [k]$$

*and*

$$\Pr_{\mathcal{M}} \left[ \sum_{l=1}^{n} x_l[\mathcal{M}(S)[i]] \geq \sum_{l=1}^{n} x_l[(i-1)d' + j] - \alpha \text{ for all } i,j \in [k] \times [d'] \right] \geq 2/3$$

*for some $\alpha \leq n/20$. Then, we have the following:*

- $\alpha = \Omega\left(\frac{k \log d'}{\varepsilon}\right)$ *if $\delta = 0$.*

- $\alpha = \Omega\left(\frac{\sqrt{k \log(1/\delta)} \log d'}{\varepsilon}\right)$ *if $\delta > 0$ and $\delta = o(1/n^2)$.*

**Theorem B.4.** *Let $d, T \in \mathbb{N}$ be sufficiently large. Any $(\varepsilon, \delta)$-differentially private algorithm $\mathcal{M}$ solving MaxSelect or MinSelect in the non-adaptive continual release model with relative error $\gamma$ and success probability $2/3$ must incur an additive error of*

- $\alpha = \tilde{\Omega}\left(\min\left\{\frac{1}{\varepsilon\gamma}, \frac{d}{\varepsilon}, \sqrt{\frac{T\log(1/\gamma)}{\varepsilon}}, \sqrt{\frac{T\log d}{\varepsilon}}, T\right\}\right)$ *if $\delta = 0$.*

- $\alpha = \tilde{\Omega}\left(\min\left\{\frac{\sqrt{1/\gamma}}{\varepsilon}, \frac{\sqrt{d}}{\varepsilon}, \frac{T^{1/3}\log^{2/3}(1/\gamma)}{\varepsilon^{2/3}}, \frac{T^{1/3}\log^{2/3}d}{\varepsilon^{2/3}}, T\right\}\right)$ *if $\delta > 0$ and $\delta = o(\varepsilon/T^2)$.*

*Proof.* Similar to the argument made in the proof of Theorem 4.3, we can show that the lower bound should be a non-decreasing function of $d$ and $\alpha = \Omega(T)$ when $\varepsilon T \leq 2$ for both MaxSelect and MinSelect. Thus, we assume $\varepsilon > 2/T$ in the remainder of the proof.

Assume $\gamma \leq 1/100$. We consider MaxSelect first. Let $S = (x_1, \ldots, x_n)$, where $x_i \in \{0,1\}^d$, $d = d'k$ for some $k$, and $n = \lceil 50\alpha \rceil$. For $i \in [k]$, define $v_i$ as the vector with $v_i[j] = 1$ for $j \in [(i-1)d' + 1, id']$ and $v_i[j] = 0$ for $j \notin [(i-1)d' + 1, id']$. Let $\overline{v_i} = 1 - v_i$. Construct a stream $(y_1, \ldots, y_{T'})$ for $T' = n + 4n(k-1) + 2n$ as follows:

- $y_t = x_t$ for $t \in [n]$.

- $y_{n+4n(i-1)+1} = \cdots = y_{n+4n(i-1)+2n} = v_i$ for $i \in [k]$.

- $y_{n+4n(i-1)+2n} = \cdots = y_{n+4ni} = \overline{v_i}$ for $i \in [k-1]$.

Note that at round $t = n + 4n(i-1) + 2n$ $(i \in [k])$, we have

$$\sum_{s=1}^{t} y_s[j] = \sum_{l=1}^{n} x_l[j] + 2ni \in [2ni, 2ni+n]$$

for $j \in [(i-1)d'+1, id']$ and

$$\sum_{s=1}^{t} y_s[j] = \sum_{l=1}^{n} x_l[j] + 2n(i-1) \in [2n(i-1), 2n(i-1)+n]$$

for $j \notin [(i-1)d'+1, id']$. If $T \geq T'$ and we run $\mathcal{M}$ on $(y_1, \ldots, y_{T'})$. Then with probability $2/3$, the output $a_t$ should satisfy

$$\sum_{s=1}^{t} y_s[a_t] \geq (1-\gamma) \max_{j \in [d]} \sum_{s=1}^{t} y_s[j] - \alpha$$

$$= (1-\gamma) \max_{j \in [(i-1)d'+1, id']} \sum_{s=1}^{t} y_s[j] - \alpha$$

Suppose $d \leq 1/100\gamma$, the above is at least

$$\max_{j \in [(i-1)d'+1, id']} \sum_{s=1}^{t} y_s[j] - \gamma n(2k+1) - \alpha \geq \max_{j \in [(i-1)d'+1, id']} \sum_{s=1}^{t} y_s[j] - \gamma n(2d+1) - \alpha$$

$$\geq \max_{j \in [(i-1)d'+1, id']} \sum_{s=1}^{t} y_s[j] - n/20.$$

Note that this is greater than $2ni - n \geq \sum_{s=1}^{t} y_s[j]$ for any $j \notin [(i-1)d'+1, id']$. Therefore, we have $a_t \in [(i-1)d'+1, id']$ and hence

$$\sum_{l=1}^{n} x_l[a_t] \geq \max_{j \in [(i-1)d'+1, id']} \sum_{l=1}^{n} x_l[j] - n/20.$$

If $T' > T$, we have $\alpha = \Omega(T/k)$. Otherwise, by Lemma B.3, we have $\alpha = \Omega\left(\frac{k \log d'}{\varepsilon}\right)$ for $\delta = 0$ and $\alpha = \Omega\left(\frac{\sqrt{k} \log d'}{\varepsilon}\right)$ for $\delta > 0$. We have to select a $k$ to maximize $\Omega\left(\min\left\{T/k, k \log d'/\varepsilon\right\}\right)$ for $\delta = 0$ and $\Omega\left(\min\left\{T/k, \sqrt{k} \log d'/\varepsilon\right\}\right)$ for $\delta > 0$. This optimization problem was discussed in (Jain et al., 2023), resulting in

$$\alpha = \tilde{\Omega}\left(\min\left\{\frac{d}{\varepsilon}, \sqrt{\frac{T \log d}{\varepsilon}}, T\right\}\right)$$

for $\delta = 0$ and

$$\alpha = \tilde{\Omega}\left(\min\left\{\frac{\sqrt{d}}{\varepsilon}, \frac{T^{1/3} \log^{2/3} d}{\varepsilon^{2/3}}, T\right\}\right)$$

for $\delta > 0$. When $d > 1/100\gamma$, the lower bound can be obtained by setting $d = 1/100\gamma$ in the above results since it is non-decreasing in $d$. Summarizing all cases yields our final lower bound.

For the proof for MinSelect, we again flip the construction by setting

- $y_t = 1 - x_t$ for $t \in [n]$.

- $y_{n+4n(i-1)+1} = \cdots = y_{n+4n(i-1)+2n} = \overline{v_i}$ for $i \in [k]$.

- $y_{n+4n(i-1)+2n} = \cdots = y_{n+4ni} = v_i$ for $i \in [k-1]$.

We can similarly show that $a_t \in [(i-1)d'+1, id']$ and

$$\sum_{l=1}^{n}(1 - x_l[a_t]) \leq \min_{j \in [(i-1)d'+1, id']} \sum_{l=1}^{n}(1 - x_l[j]) + n/25$$

for $t = n + 4n(i-1) + 2n$. Note that the above is equivalent to

$$\sum_{l=1}^{n} x_l[a_t] \geq \max_{j \in [(i-1)d'+1, id']} \sum_{l=1}^{n} x_l[j] - n/25.$$

The desired bound then follows by an identical argument. $\qquad\square$

## C. Omitted Proofs

### C.1. Proof of Theorem 3.1

*Proof of Theorem 3.1.* We prove the privacy guarantee first. When $\delta = 0$, by basic composition, it suffices to verify $\sum_{i'=1}^{T} \varepsilon_{\lceil i'/K \rceil} \leq \varepsilon/2$. In fact, we have

$$
\begin{aligned}
\sum_{i'=1}^{T} \varepsilon_{\lceil i'/K \rceil} &\leq \sum_{i=1}^{\infty} K\varepsilon_i \\
&= K \sum_{i=1}^{\lceil 4/\gamma \rceil} \varepsilon_i + K \sum_{i=\lceil 4/\gamma \rceil+1}^{\infty} \varepsilon_i \\
&\leq K\lceil 4/\gamma \rceil \cdot \frac{64(\ln T + \ln(8dT/\beta))}{\alpha} + K\sum_{i=0}^{\infty} \frac{32(\ln T + \ln(8dT/\beta))}{\alpha \cdot (1+\gamma/3)^i} \\
&\leq K \cdot 5/\gamma \cdot \frac{64(\ln T + \ln(8dT/\beta))}{\alpha} + K\frac{32(\ln T + \ln(8dT/\beta))}{\alpha} \cdot \frac{4}{\gamma} \\
&= \frac{448K(\ln T + \ln(8dT/\beta))}{\alpha\gamma} \\
&= \varepsilon/2.
\end{aligned}
$$

When $\delta > 0$, by advanced composition, it suffices to verify $\frac{1}{2}\sum_{i'=1}^{T} \varepsilon_{\lceil i'/K \rceil}^2 + \sqrt{2\ln(1/\delta)\sum_{i'=1}^{T} \varepsilon_{\lceil i'/K \rceil}^2} \leq \varepsilon/2$. Since

$$
\begin{aligned}
\sum_{i'=1}^{T} \varepsilon_{\lceil i'/K \rceil}^2 &\leq \sum_{i=1}^{\infty} K\varepsilon_i^2 \\
&= K \sum_{i=1}^{\lceil 4/\gamma \rceil} \varepsilon_i^2 + K \sum_{i=\lceil 4/\gamma \rceil+1}^{\infty} \varepsilon_i^2 \\
&\leq K\lceil 4/\gamma \rceil \cdot \left(\frac{64(\ln T + \ln(8dT/\beta))}{\alpha}\right)^2 + K\sum_{i=0}^{\infty} \left(\frac{32(\ln T + \ln(8dT/\beta))}{\alpha \cdot (1+\gamma/3)^i}\right)^2 \\
&\leq K \cdot 5/\gamma \cdot \left(\frac{64(\ln T + \ln(8dT/\beta))}{\alpha}\right)^2 + K\left(\frac{32(\ln T + \ln(8dT/\beta))}{\alpha}\right)^2 \cdot 3/\gamma \\
&\leq \frac{\varepsilon^2}{32\ln(1/\delta)},
\end{aligned}
$$

we have

$$\frac{1}{2}\sum_{i'=1}^{T}\varepsilon_{\lceil i'/K\rceil}^2 + \sqrt{2\ln(1/\delta)\sum_{i'=1}^{T}\varepsilon_{\lceil i'/K\rceil}^2} \le \frac{1}{2}\sqrt{\sum_{i'=1}^{T}\varepsilon_{\lceil i'/K\rceil}^2} + \sqrt{2\ln(1/\delta)\sum_{i'=1}^{T}\varepsilon_{\lceil i'/K\rceil}^2}$$

$$\le 2\sqrt{2\ln(1/\delta)\sum_{i'=1}^{T}\varepsilon_{\lceil i'/K\rceil}^2}$$

$$\le \varepsilon/2.$$

We then prove the utility guarantee. By Theorem 2.8, with probability $1 - \beta/4T$ the $i'$-th execution of AboveThreshold is $\alpha_i = \alpha_{\lceil i'/K\rceil}$-accurate. By Theorem 2.9, with probability $1 - \beta/4T$ the $i'$-th execution of the exponential mechanism has error no more than $2\ln(4dT/\beta)/\varepsilon_i = 2\ln(4dT/\beta)/\varepsilon_{\lceil i'/K\rceil}$. In what follows we condition on the event that all the above happen. By the union bound, this holds with probability $1 - \beta/2$.

Let $t_1 < t_2 < \cdots$ be the time-steps at which AboveThreshold returns $\top$. We will prove that with probability $1 - \beta/4$, it holds that

$$\min_{j\in[d]}\sum_{s=1}^{t_{iK}}x_s[j] \ge i\alpha/4$$

for all $i \le \lceil 4/\gamma\rceil$ if $t_{iK}$ exists.

Note that the above holds for $i = 0$ (define $t_0 = 0$). Now, suppose it holds for some $i < \lceil 4/\gamma\rceil$. Let $\tilde{t}$ be the last time-step at which

$$\min_{j\in[d]}\sum_{s=1}^{\tilde{t}}x_s[j] < (i+1)\alpha/4.$$

Our goal is to show $t_{(i+1)K} > \tilde{t}$ if $t_{(i+1)K}$ exists. This clearly holds if $\tilde{t} \le t_{iK}$. Now suppose $\tilde{t} > t_{iK}$. We will show that with probability $1 - \beta/(4 \cdot \lceil 4/\gamma\rceil)$, in the time interval $[t_{iK} + 1, \tilde{t}]$ the number of times AboveThreshold responses $\top$ is less than $K$. As a consequence, we have $t_{(i+1)K} > \tilde{t}$ if $t_{(i+1)K}$ exists.

Define the potential function at round $t$ as $\Phi_t = \sum_{j\in[d]}\exp(-\varepsilon_{i+1}\ell_t(j)/2)$, we have

$$\Phi_{t_{iK}} \le d\exp(-\varepsilon_{i+1}(i\alpha/4)/2) = d\exp(-\varepsilon_{i+1}\cdot i\alpha/8)$$

and

$$\Phi_{\tilde{t}} > \exp(-\varepsilon_{i+1}((i+1)\alpha/4)/2) = \exp(-\varepsilon_{i+1}(i+1)\alpha/8).$$

As a consequence, we have $\Phi_{t_{iK}}/\Phi_{\tilde{t}} < d\exp(\varepsilon_{i+1}\alpha/8)$. Let $s_k \in [t_{iK}+1, \tilde{t}]$ be the last time-step such that $\Phi_{s_k} \ge \Phi_{t_{iK}}/e^k$ for $k = 1, \ldots, r$, where $r \le \ln d + \varepsilon_{i+1}\alpha/8 \le \ln d + 8(\ln T + \ln(8dT/\beta))$. Assume at round $t \in [s_{k-1} + 1, s_k]$ (define $s_0 = t_{iK}$ for simplicity) AboveThreshold returns $\sigma_t = \top$, the probability that $\sigma_{t'} = \top$ for some $t' \in [t+1, s_k]$ is at most the probability that we choose some $j \in [d]$ at round $t$ such that

$$\ell_{s_k}(j) \ge \ell_{t'}(j)$$

$$\ge (1+\gamma)\cdot\min_{j'\in[d]}\sum_{s=1}^{t'}x_s[j'] + \alpha - 2\alpha_{i+1}$$

$$\ge (1+\gamma)\cdot\min_{j'\in[d]}\sum_{s=1}^{t_{iK}}x_s[j'] + \alpha - 2\alpha_{i+1}.$$

because $\sigma_{t'} = \top$ implies $q_{t'} \geq -\alpha_{i+1}$. Therefore, this probability can be bounded by

$$\sum_{j\in[d]} \frac{\exp(-\varepsilon_{i+1}\ell_t(j)/2)}{\Phi_t} \cdot \mathbb{I}[\ell_{s_k}(j) \geq (1+\gamma) \cdot \min_{j'\in[d]} \sum_{s=1}^{t_{iK}} x_s[j'] + \alpha - 2\alpha_{i+1}]$$

$$\leq \sum_{j\in[d]} \frac{\exp(-\varepsilon_{i+1}\ell_t(j)/2)}{\Phi_t} \cdot \mathbb{I}[\ell_{s_k}(j) \geq i\alpha/4 + \alpha - 2\alpha_{i+1}]$$

$$\leq \sum_{j\in[d]} \frac{\exp(-\varepsilon_{i+1}\ell_t(j)/2)}{\Phi_t} \cdot \mathbb{I}[\ell_{s_k}(j) \geq (i+3)\alpha/4]$$

since $2\alpha_{i+1} \leq \alpha/4$. Denote the above by $P$ and let $S = \{j : \ell_{s_k}(j) \geq (i+3)\alpha/4\}$, we have

$$\Phi_{s_k} = \sum_{j\in[d]} \exp(-\varepsilon_{i+1}\ell_{s_k}(j)/2)$$

$$\leq \sum_{j\in S} \exp(-\varepsilon_{i+1}((i+3)\alpha/4)/2) + \sum_{j\in[d]\setminus S} \exp(-\varepsilon_{i+1}\ell_t(j)/2)$$

$$\leq d\exp(-\varepsilon_{i+1}(i+3)\alpha/8) + (1-P)\Phi_t.$$

Rearranging the above inequality yields

$$P \leq 1 + \frac{d\exp(-\varepsilon_{i+1}(i+3)\alpha/8)}{\Phi_t} - \frac{\Phi_{s_k}}{\Phi_t}$$

$$\leq 1 + 1/e^2 - 1/e$$

$$< 4/5.$$

because

$$\Phi_{s_k}/\Phi_t \geq \frac{\Phi_{t_{iK}/e^k}}{\Phi_{s_{k-1}+1}} \geq \frac{\Phi_{t_{iK}/e^k}}{\Phi_{t_{iK}/e^{k-1}}} \geq 1/e$$

and

$$\frac{d\exp(-\varepsilon_{i+1}(i+3)\alpha/8)}{\Phi_t} \leq \frac{d\exp(-\varepsilon_{i+1}(i+3)\alpha/8)}{\exp(-\varepsilon_{i+1}(i+1)\alpha/8)}$$

$$= d\exp(-\varepsilon_{i+1}\alpha/4)$$

$$\leq 1/e^2.$$

Therefore, the probability that AboveThreshold returns at least $K$ $\top$'s during $[t_{iK}+1, \tilde{t}]$ can be bounded by the probability that the sum of $K-1$ Bernoulli random variables with mean $1/5$ is less than $r$. By the Chernoff bound (Chernoff, 1952), setting $K - 1 \geq 10r + 40\ln(4 \cdot \lceil 4/\gamma \rceil/\beta)$ ensures that this probability is at most $\beta/(4 \cdot \lceil 4/\gamma \rceil)$. Our desired conclusion follows by induction and the union bound.

Further condition on the previous result, we then use a similar argument to prove that

$$\min_{j\in[d]} \sum_{s=1}^{t_{iK}} x_s[j] \geq \alpha/\gamma \cdot (1+\gamma/3)^{i-\lceil 4/\gamma \rceil}$$

for $i \geq \lceil 4/\gamma \rceil$ (if $t_{iK}$ exists) with probability $1 - \beta/4$. We have shown that this holds for $i = \lceil 4/\gamma \rceil$. Now, assume it holds for some $i \geq \lceil 4/\gamma \rceil$ and let $\tilde{t}$ be the last time-step at which

$$\min_{j\in[d]} \sum_{s=1}^{\tilde{t}} x_s[j] < \alpha/\gamma \cdot (1+\gamma/3)^{i+1-\lceil 4/\gamma \rceil}.$$

We have

$$\Phi_{t_{iK}} \leq d\exp(-\varepsilon_{i+1}\alpha/\gamma \cdot (1+\gamma/3)^{i-\lceil 4/\gamma \rceil}/2)$$

and
$$\Phi_{\tilde{t}} > \exp(-\varepsilon_{i+1}\alpha/\gamma \cdot (1+\gamma/3)^{i+1-\lceil 4/\gamma \rceil}/2).$$

Therefore,
$$\Phi_{t_{iK}}/\Phi_{\tilde{t}} < d\exp(\varepsilon_{i+1}\alpha/\gamma \cdot (1+\gamma/3-1)(1+\gamma/3)^{i-\lceil 4/\gamma \rceil}/2)$$
$$= d\exp(\varepsilon_{i+1}\alpha(1+\gamma/3)^{i-\lceil 4/\gamma \rceil}/6).$$

Again let $s_k \in [t_{iK}+1, \tilde{t}]$ be the last time-step such that $\Phi_{s_k} \geq \Phi_{t_{iK}}/e^k$ for $k=1,\ldots,r$, where $r \leq \ln d + \varepsilon_{i+1}\alpha(1+\gamma/3)^{i-\lceil 4/\gamma \rceil}/6 \leq \ln d + 8(\ln T + \ln(8dT/\beta))$. Assume at round $t \in [s_{k-1}+1, s_k]$ AboveThreshold returns $\sigma_t = \top$, the probability that $\sigma_{t'} = \top$ for some $t' \in [t+1, s_k]$ is at most

$$\sum_{j \in [d]} \frac{\exp(-\varepsilon_{i+1}\ell_t(j)/2)}{\Phi_t} \cdot \mathbb{I}[\ell_{s_k}(j) \geq (1+\gamma) \cdot \min_{j' \in [d]} \sum_{s=1}^{t_{iK}} x_s[j'] + \alpha - 2\alpha_{i+1}]$$
$$\leq \sum_{j \in [d]} \frac{\exp(-\varepsilon_{i+1}\ell_t(j)/2)}{\Phi_t} \cdot \mathbb{I}[\ell_{s_k}(j) \geq (1+\gamma)\alpha/\gamma \cdot (1+\gamma/3)^{i-\lceil 4/\gamma \rceil} + \alpha - 2\alpha_{i+1}]$$
$$\leq \sum_{j \in [d]} \frac{\exp(-\varepsilon_{i+1}\ell_t(j)/2)}{\Phi_t} \cdot \mathbb{I}[\ell_{s_k}(j) \geq (1+\gamma/2)\alpha/\gamma \cdot (1+\gamma/3)^{i-\lceil 4/\gamma \rceil}]$$

since $2\alpha_{i+1} \leq (\gamma/2) \cdot \alpha/\gamma \cdot (1+\gamma/3)^{i-\lceil 4/\gamma \rceil}$. Denote the above quantity by $P$ and let $S = \{j : \ell_{s_k}(j) \geq (1+\gamma/2)\alpha/\gamma \cdot (1+\gamma/3)^{i-\lceil 4/\gamma \rceil}\}$, we have

$$\Phi_{s_k} = \sum_{j \in [d]} \exp(-\varepsilon_{i+1}\ell_{s_k}(j)/2)$$
$$\leq \sum_{j \in S} \exp(-\varepsilon_{i+1}(1+\gamma/2)\alpha/\gamma \cdot (1+\gamma/3)^{i-\lceil 4/\gamma \rceil}/2) + \sum_{j \in [d]\setminus S} \exp(-\varepsilon_{i+1}\ell_t(j)/2)$$
$$\leq d\exp(-\varepsilon_{i+1}(1+\gamma/2)\alpha/\gamma \cdot (1+\gamma/3)^{i-\lceil 4/\gamma \rceil}/2) + (1-P)\Phi_t.$$

Rearranging the above gives

$$P \leq 1 + \frac{d\exp(-\varepsilon_{i+1}(1+\gamma/2)\alpha/\gamma \cdot (1+\gamma/3)^{i-\lceil 4/\gamma \rceil}/2)}{\Phi_t} - \frac{\Phi_{s_k}}{\Phi_t}$$
$$\leq 1 + 1/e^2 - 1/e$$
$$< 4/5$$

because $\Phi_{s_k}/\Phi_t \geq 1/e$ and

$$\frac{d\exp(-\varepsilon_{i+1}(1+\gamma/2)\alpha/\gamma \cdot (1+\gamma/3)^{i-\lceil 4/\gamma \rceil}/2)}{\Phi_t}$$
$$\leq \frac{d\exp(-\varepsilon_{i+1}(1+\gamma/2)\alpha/\gamma \cdot (1+\gamma/3)^{i-\lceil 4/\gamma \rceil}/2)}{\exp(-\varepsilon_{i+1}\alpha/\gamma \cdot (1+\gamma/3)^{i+1-\lceil 4/\gamma \rceil}/2)}$$
$$= d\exp(-\varepsilon_{i+1}\gamma/6 \cdot \alpha/\gamma \cdot (1+\gamma/3)^{i-\lceil 4/\gamma \rceil}/2)$$
$$\leq 1/e^2.$$

Thus, the probability that AboveThreshold returns at least $K$ $\top$'s during $[t_{iK}+1, \tilde{t}]$ is bounded by $\beta/(4 \cdot \ln T/\ln(1+\gamma/3))$ given that $K-1 \geq 10r + 40\ln(4 \cdot (\ln T/\ln(1+\gamma/3)/\beta)$. The result follows by an induction argument, the union bound, and the fact that

$$i - \lceil 4/\gamma \rceil \leq \log_{1+\gamma/3} T = \frac{\ln T}{\ln(1+\gamma/3)}.$$

We now prove the final conclusion. For round $t$ such that $\sigma_t = \bot$, by we have $q_t \le \alpha_i$. Thus,

$$\sum_{s=1}^{t} x_s[a_t] = \sum_{s=1}^{t} x_s[a_{t-1}]$$

$$\le (1+\gamma) \cdot \min_{j \in [d]} \sum_{s=1}^{t} x_s[j] + \alpha.$$

For $t$ such that $\sigma_t = \top$, we know that $t = t_{i'}$ for some $i'$ and let $i = \lceil i'/K \rceil$. If $i \le \lceil 4/\gamma \rceil$, we have

$$\sum_{s=1}^{t} x_s[a_t] \le \min_{j \in [d]} \sum_{s=1}^{t} x_s[j] + 2\ln(4dT/\beta)/\varepsilon_i$$

$$\le (1+\gamma) \cdot \min_{j \in [d]} \sum_{s=1}^{t} x_s[j] + \alpha$$

as $2\ln(4dT/\beta)/\varepsilon_i \le \alpha$. Otherwise if $i > \lceil 4/\gamma \rceil$, we have

$$\sum_{s=1}^{t} x_s[a_t] \le \min_{j \in [d]} \sum_{s=1}^{t} x_s[j] + 2\ln(4dT/\beta)/\varepsilon_i$$

$$\le (1+\gamma) \cdot \min_{j \in [d]} \sum_{s=1}^{t} x_s[j] + \alpha$$

since

$$\min_{j \in [d]} \sum_{s=1}^{t} x_s[j] \ge \min_{j \in [d]} \sum_{s=1}^{t_{(i-1)K}} x_s[j]$$

$$\ge \alpha/\gamma \cdot (1+\gamma/3)^{i-1-\lceil 4/\gamma \rceil}$$

$$\ge 1/\gamma \cdot 2\ln(4dT/\beta)/\varepsilon_i.$$

$\square$

### C.2. Proof of Lemma 4.1

Given a matrix $X$, denote by $X_{(i)}$ the matrix obtained by removing the $i$-th row of $X$. A column of $X$ is called a consensus column if all entries in this column are the same. When we say an algorithm that takes as input a matrix is differentially private, we view each row of the matrix as an item. The following lemma provides a lower bound on the sample complexity of identifying the signs of a large portion of columns under approximate DP.

**Lemma C.1** ((Talwar et al., 2015; Asi et al., 2023b)). *Let $p = 1000m^2$ and $n = m/\log m$ for sufficiently large $m$. There exists a matrix $X \in \{-1, 1\}^{(n+1) \times p}$ such that:*

- *Let $W_i$ be the set of indices of consensus columns in $X_{(i)}$. Then $|W_i| \ge 0.999p$.*

- *Any algorithm $\mathcal{A} : \{-1, 1\}^{n \times p} \to \{-1, 1\}^{p}$ such that*

$$\Pr_{\mathcal{A}} \left[ \sum_{j \in W_i} \mathbb{I}[\mathcal{A}(X_{(i)})[j] = X_{1,j}] \ge 3p/4 \right] \ge 2/3$$

*for all $i \in [n+1]$ is not $(1, o(1/n^2))$-differentially private.*

We use the above lemma to prove Lemma 4.1.

*Proof of Lemma 4.1.* Let $X' \in \{-1, +1\}^{(n'+1) \times p}$ be the matrix in Lemma C.1 with $p = d$. We have $n' = \Theta(\sqrt{d}/\log d)$. Let $k = \lfloor 1/\varepsilon \rfloor$. For every $i \in [n' + 1]$, we construct a matrix $X_{(i)} \in \{-1, +1\}^{n \times d}$ by concatenating $k$ copies of $X'_{(i)}$. Consider an algorithm $\mathcal{B} : \{-1, +1\}^{n' \times d} \to \{-1, 1\}^d$ that first concatenates $k$ copies of the input matrix and then runs $\mathcal{A}$ on the resulting matrix. Note that $X'_{(i)}$ and $X_{(i)}$ have the same set of consensus columns $W_i$. We have

$$\sum_{j \in W_i} \mathbb{I}[\mathcal{B}(X'_{(i)})[j] = X'_{1,j}] = \sum_{j \in W_i} \mathbb{I}[\mathcal{A}(X_{(i)})[j] = X_{1,j}].$$

By Lemma C.1, $\mathcal{B}$ is not $(1, o(1/n^2))$-differentially private. Now, suppose $\mathcal{A}$ is $(\varepsilon, \delta)$-differentially private for $\delta = o(1/n^3)$. Group privacy implies that $\mathcal{B}$ is $(k\varepsilon, \frac{e^{k\varepsilon}-1}{e^\varepsilon -1}\delta)$-differentially private. However, we have $k\varepsilon \leq 1$ and $\frac{e^{k\varepsilon}-1}{e^\varepsilon -1}\delta = O(k\delta) = o(1/n^2)$, a contradiction. $\square$

## C.3. Proof of Lemma 4.2

*Proof of Lemma 4.2.* By standard constructions of error correcting codes, there exists $m = 2^{\Theta(d)}$ vectors $v_1, \ldots, v_m$ in $\{-1, 1\}^d$ such that

$$\mathrm{dis}(v_i, v_j) = \sum_{k \in [d]} \mathbb{I}[v_i[k] \neq v_j[k]] \geq 9d/20.$$

For every $i \in [m]$, construct a matrix $X_i \in \{-1, 1\}^{n \times d}$ by stacking $n$ copies of $v_i$. Thus, all columns in $X_i$ are consensus columns. By the utility guarantee of $\mathcal{A}$, it holds with probability at least $2/3$ that $\mathrm{dis}(\mathcal{A}(X_i), v_i) \leq d/5$, which further implies that

$$\mathrm{dis}(\mathcal{A}(X_i), v_j) \geq \mathrm{dis}(v_i, v_j) - \mathrm{dis}(A(X_i), v_i) \geq 9d/20 - d/5 = d/4$$

by the triangle inequality. Define $O_i = \{v \in \{-1, 1\}^d : \mathrm{dis}(v, v_i) \leq d/5\}$. We have $\Pr[\mathcal{A}(X_i) \in O_i] \geq 2/3$.

Note that $O_1, \ldots, O_m$ are disjoint, we have

$$1 \geq \sum_{i \in [m]} \Pr[\mathcal{A}(X_1) \in O_i]$$
$$\geq e^{-\varepsilon n} \sum_{i \in [m]} \Pr[\mathcal{A}(X_i) \in O_i]$$
$$\geq e^{-\varepsilon n} \cdot m \cdot 2/3,$$

where the second inequality uses the group privacy property of pure DP. Rearranging the above gives $n \geq \ln(2m/3)/\varepsilon = \Omega(d/\varepsilon)$. $\square$

## D. Discussion on Negative Inputs

If the inputs can take values in $[-1, 1]$, we can show that the additive error lower bound proved by Jain et al. (2023) continues to hold even when relative error is permitted. In their construction, the following operation is performed repeatedly:

- Append multiple copies of some vector $v_i \in [0, 1]^d$.

- Append the same number of copies of $\overline{v_i} = 1 - v_i$.

This operation increases the cumulative sum along every attribute. However, when inputs can be negative, we can instead use $\overline{v_i} = -v_i$. This modification ensures that the quantity of interest remains small over time. Hence, the relative error term is always negligible and their additive error bound still applies.

