# OpenReview forum: "Differentially Private Continual Release with Relative Error"
_ICML.cc/2026/Conference — ICML 2026 regular_

### Official Review · Reviewer_tb1a · 2026-03-01

**Soundness:** 3
**Presentation:** 4
**Significance:** 3
**Originality:** 3
**Overall Recommendation:** 5
**Confidence:** 3

**Summary:**

The authors consider four problems in the continual release model of differential privacy, namely MaxSum, MaxSelect, MinSum, and MinSelect. Prior work showed that high additive error is required for all these problems. In this work, the authors show that smaller additive error is possible if we allow for relative error. For the MinSelect problem, the authors show a separation between the additive error required when the input stream is fixed or chosen adaptively.

**Compliance With Llm Reviewing Policy:**

Affirmed.

**Key Questions For Authors:**

Do you have any intuition on how to close the polylog gaps between your upper and lower bounds? Do you need stronger upper bounds, lower bounds, or both?

**Limitations:**

The constants might be too large for many practical settings. This is not a significant issue since the focus of this work is to improve asymptotics.

**Strengths And Weaknesses:**

The paper is very well written and this is a nice result.

The approach for MaxSum, MaxSelect, and MinSum is relatively straightforward and easy to follow. The algorithm only updates the output occasionally. The authors use the AboveThreshold technique to detect when the error of the current output has changed significantly. The key observation is that the error is only significant at most $O(\log_{1 + \gamma}(T))$ times for relative error $\gamma$, since the true value is bounded by $T$ and can only increase.

The MinSelect problem is more complicated, since the value for the optimal attribute need not increase when the value of the currently selected item increases. This problem requires a more complicated proof for the utility guarantees for fixed input streams.
When the stream is chosen adaptively, the adversary can increase only the value of the selected element. The authors exploit this property in their lower bound to show that allowing for relative error does not improve the additive error for MinSelect in adaptive streams.

---

> ### Author Rebuttal · Authors · 2026-03-30
>
> We thank the reviewer for the insightful questions and constructive feedback. Please see our response below.
>
> > Do you have any intuition on how to close the polylog gaps between your upper and lower bounds? Do you need stronger upper bounds, lower bounds, or both?
>
> MaxSum, MinSum, MaxSelect: We are not certain if our upper bounds can be further improved. Our algorithms are relatively straightforward and we have already used decreasing privacy parameters to save a $\log T$ factor. We suspect that the lower bounds are the more promising direction for improvement. The current proofs rely on reductions from lower bounds in the batch setting. New techniques that directly analyze the sequential setting without reduction may give tighter lower bounds.
>
> MinSelect (non-adaptive setting): We believe both upper and lower bounds need to be tightened. In our algorithm, we partition the entire process into multiple stages and treat each stage separately. An algorithm that operates on the entire sequence without such partitioning could potentially achieve a better error bound. The proof of the lower bound is identical to that of MaxSelect. For the same reason, we do hope it can be improved by new analysis in the sequential setting.
>
> MinSelect (adaptive setting): As with other problems, our lower bound currently relies on a reduction from the batch setting. A better lower bound may be obtained if one could analyze the sequential setting directly.
>
> In some parameter regimes (and MinSelect in the adaptive setting), one should run the algorithms in [1] (with additive-only error). We believe some of their upper bounds can also be improved. For example, one of their algorithms relies on private sum release, which currently has no tight error bounds.
>
> [1] Jain, P., Raskhodnikova, S., Sivakumar, S., and Smith, A. The price of differential privacy under continual observation. ICML 2023.

---

> > ### Author Rebuttal · Reviewer_tb1a · 2026-04-01
> >
> > Thank you for the response. I remain positive about this submission, and I will keep my score.

---

### Official Review · Reviewer_i69H · 2026-03-09

**Soundness:** 3
**Presentation:** 3
**Significance:** 3
**Originality:** 3
**Overall Recommendation:** 5
**Confidence:** 3

**Summary:**

This paper studies relative error upper and lower bounds for DP algorithms for estimating the max/min sum or max/min index of vectors in a continual release/stream, the stream can be adaptive or non-adaptive. For max/min sum and max select problems, unified bounds for both adaptive and non-adaptive streams are given, and for min select the bounds for adaptive and non-adaptive are different. While purely additive error bounds have been established before, relative error bounds are not studied. The generic algorithm is to use above thread together with exponential mechanism: given any update, check whether the result is still accurate using above threshold, if not update the solution with exponential mechanism. The privacy parameter needs to set proportional to the number of updates, and for max/min sum and max select, this is fairly easy: after each update, the true answer is multiplied by a factor at most $1+\Theta(\gamma)$, given a sequence of length $T$, the number of updates is at most $\log T/\gamma$. This exploits the fact that these functions are monotone, and this property no longer holds for min select. To address this issue, this paper proves that after ${\rm poly}\log(d, T)$ updates, the min sum still grows by a fraction of $\Theta(\gamma)$ via a potential function argument. An adaptive lower bound for min select is also provided by reducing the problem of privately determining the signs of consensus columns in a matrix.

**Compliance With Llm Reviewing Policy:**

Affirmed.

**Final Justification:**

My positive rating remains unchanged after the authors responses.

**Key Questions For Authors:**

1. Can you comment on the requirement $\delta=o(\epsilon/T^3)$ in Theorem 4.3?

2. Would it be possible to develop an adaptive algorithm based on the non-adaptive one? Note that if the goal is to only develop adaptive algorithm *without* differential privacy, it is known how to do so [1, 2], do you think one could achieve similar results or tradeoffs under further DP constraints?

References

[1] Adversarially Robust Streaming Algorithms via Differential Privacy. JACM 2022.

[2] A framework for adversarially robust streaming algorithms. JACM 2022.

**Limitations:**

Yes.

**Strengths And Weaknesses:**

Strengths:

1. This paper studies important problems in DP in the continual release/streaming model with relative error, under adaptive and non-adaptive setting. For max/min sum and max select, it provides algorithms with bounds hold in both adaptive and non-adaptive setting, this is interesting as with additive error, a gap usually exists and a vast number of studies focus on how to reduce an adaptive problem to the non-adaptive setting with some incurred losses.

2. For min select, it shows explicit gap between adaptive and non-adaptive settings, and gives a nearly tight $\widetilde \Omega(\sqrt{d}/\epsilon)$ lower bound in the adaptive setting. The algorithm for non-adaptive setting uses a simple but neat potential function to bound the privacy budget. The lower bound argument is clean. Overall I like the theory results in this paper.

Weaknesses:

1. The paper does not provide an adaptive algorithm for min select, it would be interesting to see what type of algorithm one could design for adaptive setting, or whether one could reduce the adaptive setting to the non-adaptive algorithm.

---

> ### Author Rebuttal · Authors · 2026-03-30
>
> We thank the reviewer for the insightful questions and constructive feedback. Please see our response below.
>
> > Can you comment on the requirement $\delta=o(\varepsilon / T^3)$ in Theorem 4.3?
>
> The requirement $\delta = o(\varepsilon / T^3)$ arises from two components: the lower bound for privately determining consensus columns (Lemma 4.1), which requires $\delta = o(1/T^3)$, and an argument for proving $\alpha =\Omega(T)$ when $\varepsilon T\le 2$, which requires $\delta = o(\varepsilon / T)$.
>
> Since most existing DP techniques incur logarithmic dependence on $1/\delta$, a lower bound with $\delta = o(1/poly(T))$ is generally recognized as a hardness result. See, for example, the following references cited in the paper:
>
> Talwar, K., Guha Thakurta, A., and Zhang, L. Nearly optimal private lasso. NeurIPS 2015.
>
> Lyu, X. and Talwar, K. Fingerprinting codes meet geometry: Improved lower bounds for private query release and adaptive data analysis. STOC 2025.
>
> Dwork, C., Naor, M., Pitassi, T., and Rothblum, G. N. Differential privacy under continual observation. STOC 2010.
>
>
> > The paper does not provide an adaptive algorithm for min select
>
>
> We do not explicitly provide an adaptive algorithm for MinSelect because our lower bound (Theorem 4.3) already matches the additive-only upper bound from [3], which holds for adaptive inputs. We will clarify this point in the revision.
>
>
> > Would it be possible to develop an adaptive algorithm based on the non-adaptive one? Note that if the goal is to only develop adaptive algorithm without differential privacy, it is known how to do so [1, 2], do you think one could achieve similar results or tradeoffs under further DP constraints?
>
> While it is possible to design an adaptive algorithm using the framework of [2], the resulting error does not compare favorably to [3]. This is because [2] requires running multiple copies of non-adaptive algorithms (specifically, a number called the "flip number"), which consumes much more privacy budget. In the problem we consider (MinSelect), the flip number can be as large as $T$, leading to an $O(T)$ error under pure DP and $O(\sqrt{T})$ error under approximate DP. Both of them are larger than the bounds in [3] ($O(\sqrt{T})$ and $O(T^{1/3})$, respectively).
>
> Although [1] requires fewer copies, their framework aggregates results from the copies at each round privately. However, the output of MinSelect is simply an index. It is unclear how to privately aggregate multiple indices in this problem.
>
> References
>
> [1] Adversarially Robust Streaming Algorithms via Differential Privacy. JACM 2022.
>
> [2] A framework for adversarially robust streaming algorithms. JACM 2022.
>
> [3] Jain, P., Raskhodnikova, S., Sivakumar, S., and Smith, A. The price of differential privacy under continual observation. ICML 2023.

---

> > ### Author Rebuttal · Reviewer_i69H · 2026-03-31
> >
> > I thank the authors for the responses, they have addressed my questions. As my initial rating is positive, I'll keep the score as is.

---

### Official Review · Reviewer_k2qn · 2026-03-11

**Soundness:** 4
**Presentation:** 3
**Significance:** 3
**Originality:** 4
**Overall Recommendation:** 5
**Confidence:** 2

**Summary:**

This paper investigates the problem of differentially private continual release for four fundamental tasks: MaxSum, MinSum, MaxSelect, and MinSelect. Prior research established that algorithms for these tasks suffer from a large purely additive error. The authors demonstrate that introducing a relative error allowance can significantly reduce the additive error for most of these tasks. Specifically, they develop improved algorithms that achieve these smaller error bounds for MaxSum, MinSum, and MaxSelect under both nonadaptive and adaptive streams, as well as for MinSelect under nonadaptive streams. A key contribution of this work is establishing a strict separation between nonadaptive and adaptive streams. The authors prove that for the MinSelect task under adaptively generated streams, incorporating a relative error fails to improve the additive error, meaning a large error remains unavoidable.

**Compliance With Llm Reviewing Policy:**

Affirmed.

**Final Justification:**

Overall, the response significantly improved the paper’s clarity, and I will increase my score from 4 to 5.

**Key Questions For Authors:**

1. Regarding Algorithm 2 and the decreasing sequence of privacy parameters, how does this fragmentation of the privacy budget impact the hidden constant factors in the polylogarithmic bound? Providing a concrete numerical example for typical values like T=1,000,000 and d=1,000 would clarify the practical feasibility.
2. The adaptive lower bound for MinSelect relies on an adversary that specifically targets the released zero-sum attribute by repeatedly placing 1 on it. Does this worst-case theoretical adversary align with realistic adaptive scenarios? Are there intermediate weakly adaptive models where smaller error bounds might still be achievable?
3. The algorithms heavily rely on the Above Threshold mechanism to save privacy budget when the answer does not change significantly. However, continuous querying still consumes budget. For an infinitely long stream or an extremely long horizon, does the algorithm eventually run out of budget under pure differential privacy?

**Limitations:**

Yes.

**Strengths And Weaknesses:**

**Strengths:**
1. The paper is technically sound. Theoretical claims are supported by complete mathematical proofs. The use of the Above Threshold mechanism to lazily update the answers with decreasing privacy budgets is carefully analyzed using potential functions. The lower bound proofs, particularly the reduction to the adaptive MinSelect problem, are robust and convincing.
2. Introducing relative errors to bypass the lower bound of pure additive errors is a very pragmatic and valuable idea. More importantly, the paper explicitly demonstrates that there is a strict boundary between non-adaptive and adaptive flows, which contributes to the theoretical basis of differential privacy.
3. Applying relative error guarantees to these specific continual observation tasks is novel. The most original contribution is the construction of the adaptive lower bound for MinSelect. Proving that relative error cannot save the additive error in adaptive settings separates the two stream paradigms.

**Weaknesses:**
1. One limitation is that the asymptotic bounds seem to hide some significant constants or logarithmic terms. This might make the algorithm impractical for streams of moderate length. Furthermore, the MinSelect algorithm assumes that the minimum sum grows multiplicatively after a set number of updates, which relies heavily on the stream being nonadaptive.
2. However, the potential function pops up a bit abruptly in the analysis. It would really help readers from a broader machine learning background if there was a clearer explanation of why this specific exponential form captures the nonadaptive growth so well.
3. The scope of the tasks regarding simple sums and selections is fundamental. While it lays good theoretical groundwork, the direct applicability of these specific algorithms to more complex machine learning models is limited without further nontrivial extensions.

---

> ### Author Rebuttal · Authors · 2026-03-30
>
> We thank the reviewer for the insightful questions and constructive feedback. Please see our response below.
>
> > How does the decreasing sequence of privacy parameters in Algorithm 2 impact the hidden constants? Providing a concrete numerical example for typical values like T=1,000,000 and d=1,000 would clarify the practical feasibility.
>
> The setting of decreasing privacy parameters does not affect the hidden constants. These constants are determined by our analysis in the proof (Appendix C.1).
>
> In practice, parameters such as $K$ and $\alpha$ can be set manually rather than strictly following the theoretical values. For example, consider a 1,000,000 x 1,000 synthetic time-series dataset composed of sinusoidal periodic signals, linear trends, and Gaussian noise, normalized to [0,1]. The maximum difference between the maximum and minimum cumulative sums can be as large as $\approx 1600$. Running the algorithm with $K=1$ and $\alpha = 2000$ yields the following errors, which are significantly smaller than $1600$:
>
> | $\gamma$ | Additive Error  |
> | -------- | ------ |
> | 0.01     | 137.69 |
> | 0.03     | 16.36  |
> | 0.05     | 12.7   |
> | 0.1      | 4.22   |
> | 0.3      | 2.93   |
>
> > Does the worst-case theoretical adversary that specifically targets the released zero-sum attribute by repeatedly placing 1 on it align with realistic adaptive scenarios? Are there intermediate weakly adaptive models where smaller error bounds might still be achievable?
>
>
> Consider a platform that privately publishes the least crowded road. Newcomers are likely to choose this road, thereby increasing its traffic. This process aligns with an adaptive adversary.
>
> There do exist weaker models, such as smoothed analysis in online learning. In this model, there is an underlying distribution $D$, and the data at round $t$ is drawn from some distribution $D_t$ chosen by an adaptive adversary under the restriction that $D_t$ is smooth w.r.t. $D$. This model is weaker than the fully adaptive model we considered in the paper. It might be possible to obtain smaller error bounds within it.
>
> Reference: Haghtalab, N., Roughgarden, T., and Shetty, A. Smoothed analysis of online and differentially private learning. NeurIPS 2020
>
>
> > For an infinitely long stream or an extremely long horizon, does the algorithm eventually run out of budget under pure differential privacy?
>
>
> There are general strategies for converting algorithms designed for a known horizon $T$ into ones that work for extremely long (or even infinite) input streams without exhausting the privacy budget.
>
> For example, one can run multiple copies in parallel, each with horizon $T_k = 2^{2^k}$ and $\varepsilon_k = \varepsilon/2^k$ for $k = 1,2,\dots$. Since $\sum_{k=1}^{\infty}\varepsilon_k = \sum_{k=1}^{\infty}\varepsilon/2^k = \varepsilon$, we will not run out of privacy budget.
>
> At round $t$, we find the smallest $k$ such that $T_k \ge t$ and output the answer from the $k$-th copy. Since our algorithm has error $polylog(T)$, one can argue that the final error bound of the new algorithm is also $polylog(t)$.
>
> > Why use the potential function in the analysis?
>
> Our algorithm (approximately) selects the index with the minimum cumulative sum using the exponential mechanism. This is structurally identical to the multiplicative weights method used in online learning. The potential function we employ is standard in analyzing the regret of online learning. We adopt it to leverage existing proof techniques from that literature.
>
> > Discussion of practical limitations
>
> Our algorithms involve large hidden constants in the resulting error bounds and, therefore, may not be feasible in practice. This work focuses on understanding asymptotic behavior theoretically. We leave practical implementation and optimization to future work.

---

> > ### Author Rebuttal · Reviewer_k2qn · 2026-04-01
> >
> > Thank you for the clear and thoughtful rebuttal. The authors addressed my main concerns well, especially by clarifying that the decreasing privacy parameters do not affect the hidden constants, providing a concrete numerical example to illustrate practical behavior, and discussing both weaker adaptive models and the infinite-horizon setting. Overall, the response significantly improved the paper’s clarity, and I will increase my score from 4 to 5.

---

### Official Review · Reviewer_tKGr · 2026-03-17

**Soundness:** 3
**Presentation:** 4
**Significance:** 3
**Originality:** 3
**Overall Recommendation:** 5
**Confidence:** 4

**Summary:**

The paper studies differentially private continual release. The mechanism observes $d$ time series, each of length $T$, and at each time step it must privately output one of the following quantities:
- The maximum cumulative sum of any series
- The minimum cumulative sum of any series
- The index of the series with the maximum cumulative sum
- The index of the series with the minimum cumulative sum

Previous work showed that no private mechanism can achieve small additive error on every time step with high probability. This paper gives algorithms that achieve polylog(d, T) additive error, but also constant multiplicative error. The positive results are matched by nearly tight lower bounds.

The basic idea is to maintain an estimate of the quantity of interest, and use the AboveThreshold mechanism to check at each time step whether this estimate satisfies the multiplicative and additive guarantees. Due to the constant multiplicative factor, the guarantee can be violated at most $O(\log T)$ times (the elements of the time series are assumed to be bounded). And since AboveThreshold only pays a privacy cost when its check fails, the total error is polylog(d, T). While the basic idea is simple, the analysis is quite intricate.

**Compliance With Llm Reviewing Policy:**

Affirmed.

**Final Justification:**

Accept. I suggest that the authors add the text from their rebuttal which explains why MinSelect is so different from the other problems, and how the difficulty of the problems would change if the values were in the interval [-1, 1].

**Key Questions For Authors:**

- I am struggling to understand why the MinSelect problem is so different than the other 3 problems. At first glance, it seems entirely symmetric to MaxSelect. My best guess is that the values being in the interval [0, 1] is playing an important role. Since the cumulative sums are monotonically increasing, estimating the min and max require different techniques, and one problem is harder than the other. Is that correct? If so, why isn't MinSum also a hard problem? How much would the problem change if the value were in the interval [-1, 1]? I think the paper's clarity would be greatly improved by an extended discussion of this issue.

- The lower bound for MinSelect for adaptive sequences does not state the dependence of $\alpha$ (the additive error) on $\gamma$ (the multiplicative error). What is this dependence? Is there any non-trivial value of $\gamma$ for which $\alpha$ can be zero? In other words, is it possible to give a purely multiplicative error guarantee? (I'm addressing this question to all 4 problems, not just MinSelect.)

**Limitations:**

Yes

**Strengths And Weaknesses:**

I think that relaxing the error requirements to be both multiplicative and additive is quite reasonable, especially since it leads to circumvention of previous lower bounds. This new framing of the problem yields an important insight. Overall, I think the paper is a meaningful contribution to the literature on DP continual release.

---

> ### Author Rebuttal · Authors · 2026-03-30
>
> We thank the reviewer for the insightful questions and constructive feedback. Please see our response below.
>
> > Why MinSelect is so different from other problems?
>
> Our algorithms for other problems rely on a key property: once the previous estimate becomes erroneous, the quantity of interest grows by $1+O(\gamma)$ . However, this does **not** hold for MinSelect.
>
> When the current output becomes inaccurate, we have:
>
> - MaxSelect: the maximum cumulative sum is $1+O(\gamma)$ times larger than that of the current output.
> - MinSum: the minimum cumulative sum is at least $1+O(\gamma)$ times the current output.
> - MinSelect: the cumulative sum of the current output is at least $1+O(\gamma)$ times the minimum cumulative sum.
>
> For the MaxSelect and MinSum, the maximum/minimum cumulative sum grows by $1+O(\gamma)$ after each update. However, for MinSelect, we only know that the sum of the current output grows by $1+O(\gamma)$, while the minimum cumulative sum may remain unchanged.
>
> > How much would the problem change if the value were in the interval [-1, 1]?
>
> If the values were restricted to [-1, 1], our algorithms would fail because the number of updates could no longer be bounded by $O(\log T)$. In fact, a slight modification of the construction from [1] yields the same lower bounds as in the purely additive case. Thus, allowing relative error does not lead to improved error bounds. We will provide a more detailed discussion in the revision.
>
> > What is the dependence of $\alpha$ (the additive error) on $\gamma$ (the relative error) in the lower bound for MinSelect for adaptive sequences?
>
> Our lower bound holds for any $\gamma \ge 0$ and matches the purely-additive (i.e., $\gamma = 0$) upper bound from [1]. Hence, there is **no** dependence of $\alpha$ on $\gamma$. We will clarify this in the revision.
>
> The reason is that our construction always contains a zero-sum attribute. As a consequence, for any $\gamma \ge 0$ the relative error term $\gamma \cdot 0 = 0$ does not affect the result.
>
> > Is it possible to give a purely multiplicative error guarantee?
>
> No, a purely multiplicative error guarantee is impossible --- a positive additive error term is always necessary.
>
> Consider MaxSum with all input values equal to $0$. The true answer is $0$. To satisfy DP, any private algorithm must incur some additive error $\alpha > 0$, even if $\gamma$ can be arbitrarily large, since $0\cdot \gamma = 0$ for any $\gamma$.
>
> References
>
> [1] Jain, P., Raskhodnikova, S., Sivakumar, S., and Smith, A. The price of differential privacy under continual observation. ICML 2023.

---

> > ### Author Rebuttal · Reviewer_tKGr · 2026-04-01
> >
> > Thanks for the clarifications! I already thought that the paper should be accepted, so I will leave my score unchanged.

---

### Decision · Program_Chairs · 2026-04-30

**Decision:**

Accept (regular)

**Comment:**

The papers studied the continual release model in DP and studies several core tasks in this regard. All reviewers found the paper to be interesting, novel and dealing with a core ML problem under DP, and advocated acceptance to ICML. We have no choice but to agree -- terrific work! We recommend acceptance to ICML.
We also encourage the authors to address the reviewers' feedback in the final version of the paper, regarding presentation issues and references.